# RPC-Bench: A Fine-grained Benchmark for Research Paper Comprehension

## Abstract

Leveraging large foundation models for document understanding has emerged as a rapidly advancing research area. Unlike general-purpose documents, research papers constitute a particularly challenging domain, as they are characterized by complex figures, detailed tables, and highly specialized scientific knowledge. However, existing benchmarks pay limited attention to evaluating the fine-grained capabilities of current models in comprehending research papers at scale. To address this gap, we propose RPC-Bench, a large-scale fine-grained question-answering benchmark constructed from review-rebuttal exchanges of high-quality academic papers, with each paper available in two input formats (pure text and rendered page images) enabling evaluation of both large language models (LLMs) and visual language models (VLMs). We design a fine-grained taxonomy aligned with the research flow of academic papers to guide annotation. We also define an elaborate LLM–human interaction annotation framework to support large-scale labeling and quality control. Following the LLM-as-a-Judge paradigm, we develop a scalable framework that evaluates models on correctness-completeness and conciseness, with high agreement to human judgment. Experiments show GPT-5 leads with a 66.54% correctness-completeness score, dropping to 35.05% after conciseness adjustment. In addition, multimodal LLMs perform better on pure text than visual–text inputs, highlighting the need for improved visual integration in scholarly document understanding. Our code and data is available[1].

## 1 Introduction

Large foundation models are increasingly serving as research copilots, supporting tasks such as deep research (Schmidgall et al., 2025; Cappello et al., 2025), paper reviewing (Chang et al., 2025; Zhang et al., 2025), and even the automation of the research process itself. A critical prerequisite for these applications is the ability of both large language models (LLMs) and vision–language models (VLMs) to achieve a comprehensive understanding of research papers. Beyond parsing explicit content, such models must be able to grasp specialized concepts, analyze methodological motivations, and evaluate experimental limitations that can guide subsequent scientific discovery.

Although document understanding has made remarkable progress in recent years, existing works have largely focused on general-domain tasks such as layout analysis, content localization, and caption generation. In contrast, research paper understanding is substantially more complex: it demands domain-specific expertise to interpret technical concept, specialized methodologies, and detailed experimental designs. For example, PeerQA (Baumgärtner et al., 2025) is restricted in scale, covering only a small number of question-answer (QA) pairs. SPIQA (Pramanick et al., 2024), DocGenome (Xia et al., 2024), and ArXivQA (Li et al., 2024b) rely heavily on synthetic QA rather than real scholarly interactions, limiting their authenticity. More broadly, these benchmarks are constrained by task-based taxonomies without categorizing questions by the depth of content understanding. They also employ limited evaluation metrics and often lack simultaneous support for both textual and visual inputs. As shown in Table 1, there is no comprehensive benchmark that evaluates the ability of various methods to deeply understand large-scale research papers.

To address this gap, **we introduce RPC-Bench, a large-scale benchmark designed for in-depth research paper comprehension.** RPC-Bench is constructed from high-quality publications

---

[1] https://anonymous.4open.science/r/PRC-Bench-B327

Table 1: Comparison with the most relevant research paper Benchmarks. Conc.=Conciseness; Corr.=Correctness; F1-like is computed as the harmonic mean of correctness and completeness; inp.=input. "Eval. Metrics" are LLM-based metrics.

| Benchmarks | Papers | QA | Real QA | Taxonomy | Eval. Metrics | Textual inp. | Visual inp. |
|---|---|---|---|---|---|---|---|
| PeerQA | 208 | 579 | ✓ | task | Corr. | ✓ | ✗ |
| SPIQA | 25.5K | 270K | ✗ | task | LLMLogScore | ✓ | ✓ |
| ArXivQA | 16.6K | - | ✗ | task | - | ✗ | ✓ |
| DocGenome | 500K | - | ✗ | task | GPT-acc | ✗ | ✓ |
| RPC-Bench | 4050 | 46.3K | ✓ | content | Conc., F1-like | ✓ | ✓ |

(2013–2024) on OpenReview[2], together with their associated review–rebuttal exchanges from the peer-review process. Unlike synthetic datasets, our QA pairs are directly derived from these authentic review–rebuttal interactions and transformed into question–answer format through a collaborative LLM–human workflow, ensuring both diversity and reliability. After rigorous filtering, the final benchmark encompasses 4,050 papers and 46.3K QA pairs.

To systematically capture the essential aspects of paper understanding, we decompose the research workflow into a fine-grained taxonomy of three primary dimensions (Concepts, Methods, and Experiments) further divided into nine categories. This taxonomy provides principled guidance for annotation and evaluation, enabling nuanced assessment across the full spectrum of academic paper comprehension.

In addition, we design a scalable evaluation framework based on LLM judges that align with human judgment, supporting both pure-text and rendered-page inputs to benchmark LLMs and VLMs alike. Model outputs are evaluated jointly on correctness (accuracy of generated responses, akin to precision), completeness (coverage of essential content, akin to recall), and conciseness, with multiple pilot-tested LLM judges aggregated to produce stable, human-consistent scores.

We conduct extensive experiments across 19 state-of-the-art models, including 6 LLMs, 3 document-centric models (DCMs), 5 VLMs, and 5 retrieval-augmented generation (RAG) models. Our results reveal that none of the evaluated models are capable of fully comprehending research papers. Even the best model, GPT-5, achieved only 66.54% on F1-like (harmonic mean of correctness and completeness), which dropped to 35.05% under the conciseness-constrained F1-like. Furthermore, for multimodal-capable LLMs, replacing text inputs with page-image inputs consistently reduced F1-like by 4.57–33.40%, highlighting the current weakness of models in visual reasoning over scholarly documents. In summary, our contributions are:

• We introduce RPC-Bench, the first large-scale benchmark grounded in authentic review–rebuttal exchanges, featuring a fine-grained taxonomy aligned with the research workflow to enable systematic annotation and evaluation of research paper comprehension.

• We introduce a LLM–human collaborative annotation framework that supports large-scale QA generation and rigorous quality control.

• We develop a scalable evaluation framework that jointly measures the balance among correctness, completeness, and conciseness with high consistency to human judgment.

• We conduct a comprehensive empirical study on 19 advanced models, identifying fundamental limitations in both text-based and multimodal research paper understanding.

## 2 RELATED WORK

**Methodologies for Document Question Answering.** The current landscape of document QA is organized around three complementary pillars: (i) large foundation models, (ii) document-centric architectures, and (iii) retrieval-augmented generation (RAG). State-of-the-art LLMs and VLMs include proprietary models like GPT-5 (Leon, 2025), Claude 4.1, and Gemini 2.5 (Comanici et al., 2025), as well as leading open-source models such as the Qwen (Yang et al., 2025), GLM (GLM et al., 2024), and DeepSeek (Deng et al., 2025) series.

Document-centric architectures are introduced to accommodate to the structural and layout peculiarities of long documents. One line of the work, such as Monkey-Chat-7B and DocOwl2-8B (Li

---

[2]https://openreview.net/

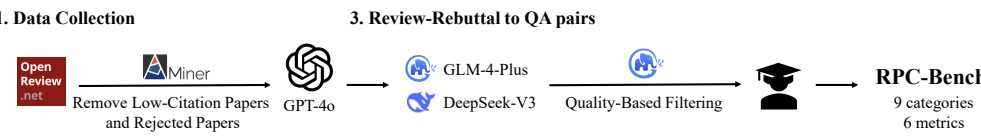

Figure 1: RPC-Bench Construction Pipeline. Papers and review–rebuttal pairs are crawled from OpenReview, low-citation and rejected papers are removed via AMiner, segmented into comment–response units with GPT-4o, rewritten into QA pairs using GLM-4-Plus and DeepSeek-V3, and low-quality QA items are discarded before iterative human annotation and review.

et al., 2024c; Hu et al., 2024), enables direct, OCR-free document understanding, avoiding error propagation from external OCR. Another line, exemplified by layout-aware models like **DocLLM** Wang et al. (2023) and **Docopilot-8B** (Duan et al., 2025), explicitly models two-dimensional page layout to better parse complex structures like tables and forms.

RAG-based approaches mitigates the limited parametric knowledge of models when operating over vast corpora by grounding generation in retrieved evidence. Textual RAG methods include **RAP-TOR** (Sarthi et al., 2024), which uses recursive clustering, as well as **HippoRAG** (Gutiérrez et al., 2025) and **MemoRAG** (Qian et al., 2025), which optimize indexing and memory. More broadly, multimodal RAG variants such as **VisRAG** (Yu et al., 2024) and **VDocRAG** (Tanaka et al., 2025) extend retrieval to visual content, enabling evidence discovery within figures and tables. The RAG ecosystem is further supported by toolkits like **FlashRAG** (Jin et al., 2025) and explorations into alternative data structures such as knowledge graphs with **GraphRAG** (Edge et al., 2025).

**Document QA Benchmarks.** Numerous benchmarks have been developed to standardize evaluation. For instance, SPIQA (Pramanick et al., 2024) focuses on multimodal questions concerning figures and tables in scientific papers. DocGenome (Xia et al., 2024) provides a large-scale, multi-domain dataset for both pre-training and high-level evaluation. LongDocURL (Deng et al., 2024) marks a step toward finer granularity by assessing the distinct skills of understanding, reasoning, and locating. PeerQA (Baumgärtner et al., 2025), akin to our work, is limited to textual data with a relatively small set of annotated QA pairs, thereby neglecting the crucial multimodal elements inherent in research papers. However, existing benchmarks suffer from several notable limitations. (1) **Limited scale**: Existing benchmarks are often small in scope, failing to cover large collections of research papers and question–answer pairs. (2) **Quality issues**: Many rely on automatically generated QA pairs whose correctness is hard to guarantee, and they usually classify tasks only by type rather than by the depth or nature of question content. (3) **Narrow and shallow evaluation**: Current benchmarks tend to emphasize multimodal QA while lacking systematic assessment of paper understanding and reasoning. Moreover, they typically focus on single-dimensional metrics such as accuracy, which cannot fully capture the quality of long-text answers.

Unlike prior works, our benchmark offers large-scale, realistic academic QA grounded in peer reviews and rebuttals. We categorize questions according to the stages of academic research and assess a broad spectrum of document QA methods. Furthermore, we transform eligible peer-review QA pairs into binary classification tasks. Our comprehensive evaluation reveals that current approaches still exhibit pronounced limitations in expert-level comprehension of scholarly literature.

## 3 RPC-BENCH

RPC-Bench is a challenging benchmark for research paper comprehension in realistic settings, designed to evaluate the ability to accurately comprehend complex domain-specific knowledge and to reason over scholarly content. (1) curating large-scale open-ended QA pairs and yes/no QA pairs from high-quality papers and their review–rebuttal exchanges; (2) introducing a fine-grained taxonomy and rigorous metrics for systematic assessment; and (3) defining a carefully designed LLM–human interactive annotation framework to enable large-scale labeling and strict quality control. The following sections detail the data collection, taxonomy design, annotation process, and evaluation protocol. Figure 1 presents the overall framework for benchmark construction.

## 3.1 DATA COLLECTION

To rigorously assess model capabilities in paper understanding, we built a three-stage data pipeline:

- Broad coverage collection: Collected 44.7K publicly available peer-reviewed papers and their corresponding review–rebuttal pairs from OpenReview[3] spanning 2013–2024.
- Quality refinement: Matched the collected papers against the popular academic search and mining system AMiner[4] to remove incomplete entries, resulting in a curated set of 17.7K papers.
- Impact-based sampling: Selected 3521 accepted papers with $\geq 50$ citations as positive samples, plus 361 highly-cited and 361 random rejected papers as challenging negatives to enhance the robustness and generalization of the benchmark.

This pipeline yields a scholarly collection of 4243 papers. We chronologically split this collection as follows: 3556 papers for training, 487 papers for validation, and 200 papers for testing.

## 3.2 TAXONOMY DESIGN

Our goal is to assess whether models truly comprehend and reason about scholarly articles, rather than relying on memorization or surface pattern-matching. To this end, we designed a taxonomy guided by the natural research flow of academic papers. It begins with ***what-questions***, which focus on clarifying fundamental concepts, terminology, and contextual background. It then advances to ***how-questions***, which explore the mechanics of methods, algorithmic details, and experimental setups. Finally, it deepens into ***why-questions***, which examine the underlying motivations of methods and the reasoning behind observed experimental outcomes. By structuring questions in this layered way—moving from foundational understanding, to technical mechanics, to deeper rationale—the taxonomy not only reveals the strengths of existing works but also highlights their limitations. These insights, in turn, help identify open gaps and inspire new directions for future research.

---

**Task Taxonomy**

1. Concept Understanding (C.U.) [`What`]: Clarifies or explains key concepts, terminology, theoretical viewpoints, or information conveyed in figures, tables, or formulas.
2. Methods
   2.1. Method Disambiguation (M.D.) [`What`]: Clarifies methodological details to resolve misunderstandings or ambiguities, ensuring an accurate grasp of proposed approaches.
   2.2. Method Mechanics (M.M.) [`How`]: Questions about the implementation or function of methodological workflow or components, such as the effect of specific modules in models.
   2.3. Motivation Analysis (M.A.) [`Why`]: Examines the rationale, principles, or intentions underlying a proposed method or decision.
   2.4. Method Comparison (M.C.): Compares the proposed approach with baseline methods, analyzing similarities, differences, or performance to highlight novelty.
3. Experiments
   3.1. Experimental Exposition (E.E.) [`What`]: Describes experimental outcomes, infers how modifications or variations could impact results or conclusions, and addresses reasoning tasks such as calculation, counting, or comparative analysis.
   3.2. Experimental Setup (E.S.) [`How`]: About the design, configuration, and execution of experiments.
   3.3. Experimental Analysis (E.A.) [`Why`]: Studies the reasons of specific experimental outcomes, links them to the proposed approach, and assesses their generalizability and potential impact.
4. Claim Verification (C.V.): Binary classification tasks that assess the correctness of claims, hypotheses, or experimental conclusions.

---

Based on this principle, we define a four-granularity taxonomy organized around key components of research papers (see taxonomy above), enabling fine-grained and multi-perspective coverage of academic paper understanding. Most categories are formulated as free-form QA tasks, while the

---

[3]https://openreview.net/
[4]https://www.aminer.cn/

Verification category is implemented as a binary classification task. Both formats require models to locate, integrate, understand, and reason over information drawn from the target paper.

### 3.3 ANNOTATION PROCESS

Manual annotation of taxonomy-based QA pairs requires domain expertise and significant time for labeling and verification, making large-scale, high-quality data collection prohibitively costly. To mitigate this, we propose a semi-automated pipeline that leverages multiple large language models (LLMs) in a hybrid manner, reducing human effort while maintaining annotation quality.

Crawled review–rebuttal pairs from OpenReview usually contain overall reviews and general replies rather than paired comment–response matches. To enable fine-grained question generation, we first use GPT-4o (chosen for its strong reasoning and contextual understanding) to decompose each review into minimal, self-contained comment–response pairs. Guided by our taxonomy, GLM-4-Plus and DeepSeek-V3 then rewrite these pairs into free-form QA or claim verification tasks, assigning each to the proper taxonomy category. This setup, validated in a pilot study, achieves competitive rewriting quality at far lower cost than GPT-4o, making it practical for large-scale generation.

Finally, to ensure the reliability of the automatically generated questions, we apply a two-stage filtering process using GLM-4-Plus: review-rebuttal filtering and QA-level filtering. The filtering criteria exclude entries lacking substantive academic content, including purely mechanical changes, reliance on external resources, indirect or unanswered replies, and superficial commitments without concrete improvements. Detailed criteria and examples are provided in Appendix C.1.

We employed four annotators (Master's degree or above), with two handling annotation and two reviewing. Before formal annotation, all annotators received training and practiced QA conversion on 10 sample papers, with iterative feedback until each reached a $\geq$95% pass rate. To ensure quality over speed, annotators were limited to 80 QA pairs per day, averaging 5–6 minutes per question. Annotated data were promptly reviewed, and problematic cases were returned for correction.

Using the annotation platform (Appendix C.2), annotators examined each segmented review–rebuttal pair and chose the better output between GLM-4-Plus and DeepSeek-V3, while verifying taxonomy labels. If both outputs were inadequate, they rewrote the pair manually and assigned the correct category. To reduce bias, model identities were anonymized as Model1 and Model2, with randomized ordering. Annotators could discard low-quality pairs or generate multiple sub-questions from a single pair, provided each addressed a distinct aspect. The review platform (Appendix C.3) displayed both original and rewritten content, allowing reviewers to approve or reject entries. Rejections required specific feedback to guide further revisions. Due to cost constraints, only the validation and test sets were manually annotated, while the training set retained QA pairs generated by LLMs. Table 2 summarizes the dataset statistics, and Figure 2 presents the distribution of data across different categories and domain. The full set of prompts used in this stage, along with the graphical user interface for annotation and review, are provided in Appendix C.

Table 2: Statistics of the RPC-Bench. A/M Q: average/max question length. A/M A: average/max answer length. Lengths are measured in words.

| Statistics | train | val | test |
|---|---|---|---|
| Papers | 3427 | 423 | 200 |
| Accept | 2218 | 314 | 116 |
| year | 2013-2022 | 2022-2023 | 2024 |
| Venue | 15 | 7 | 4 |
| QA | 39203 | 6152 | 2787 |
| A/M Q | 25.4/157 | 24.9/226 | 24.2/250 |
| A/M A | 72.4/320 | 93.9/1337 | 87.9/773 |

### 3.4 QUALITY CONTROL

Using a two-stage semi-automated pipeline combining LLM-based filtering and expert validation, we ensured annotation quality and retained only questions answerable from the paper's content, thus ensuring the RPC-Bench reliably assesses research paper comprehension.

As described in Section3.3, GLM-4-Plus was employed to remove low-quality or unanswerable data from both review–rebuttal pairs and converted QA items across the entire dataset, including the training set. This step eliminated 24.5% of review–rebuttal pairs and 12.96% of QA items, respectively.

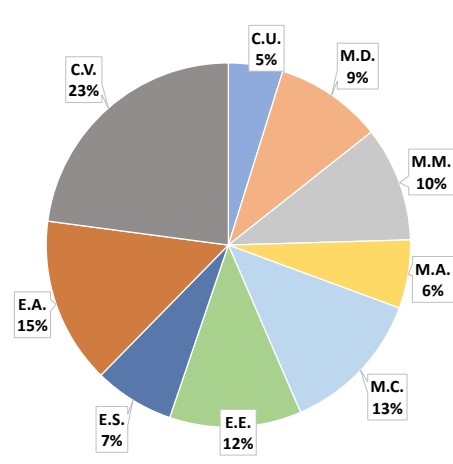 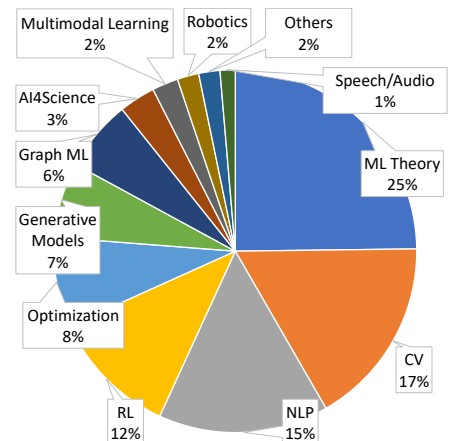

(a) Distribution of taxonomy-defined categories of RPC-Bench (val and test).

(b) Domain distribution of RPC-Bench. ML, Machine Learning; CV, Computer Vision; NLP, Natural Language Processing; RL, Reinforcement Learning.

Figure 2: Comparative overview of category and domain compositions in RPC-Bench.

During annotation, reviewers inspected the annotated data, returned problematic cases to the original annotators for correction, and jointly verified question answerability. This stage further removed 8.87% of QA items identified as low-quality or unanswerable based solely on the information in the corresponding paper. For QA items referencing specific numbers, formulas, or section indices, annotators and reviewers were required to verify their presence in the final paper version and update all indices accordingly to maintain positional accuracy and consistency.

### 3.5 EVALUATION PROTOCOL

To enable consistent and flexible assessment of academic paper understanding, we propose a standardized evaluation framework. For binary classification with clear ground-truth labels, accuracy (ACC) is used as the primary metric. For open-ended QA, traditional automatic metrics (e.g., BLEU, BERTScore) often fail to capture answer quality due to diverse valid semantically-equivalent expressions. Following recent work on LLM-as-a-Judge (D'Souza et al., 2025; Desmond et al., 2025; Li et al., 2024a), we adopt an LLM-based evaluation that scores responses along three dimensions: conciseness (brevity and without irrelevant detail), correctness (accuracy and fidelity, akin to precision), and completeness (coverage of essential content, akin to recall). Each is rated on a 0–5 scale. We further compute two derived metrics: an F1-like score (harmonic mean of correctness and completeness) and informativeness, the aggregate of all three dimensions.

$$\text{F1-like} = \frac{(1 + \beta^2) \times (\text{Correctness} \times \text{Completeness})}{\beta^2 \times \text{Correctness} + \text{Completeness}}, \quad \text{Informativeness} = \text{F1-like} \times \frac{\text{Conciseness}}{5}$$

where $\beta$ controls the weight between correctness and completeness ($\beta = 1$ by default), while conciseness is normalized to [0,1]. This captures the F1-like balance of correctness and completeness, with conciseness penalizing verbosity. Full evaluation prompts are given in Appendix C.7.

## 4 EXPERIMENTS

### 4.1 EXPERIMENTAL SETUP

We evaluate models on the RPC-Bench test set under two input settings: pure-text and image-based. For text, each PDF is converted to Markdown via MinerU[5] , with content truncated if it exceeds the model's context window. For images, PDFs are rendered with PyMuPDF at 200 DPI, and the first 15 pages are used to balance coverage and context limits. For models without multi-image support (e.g., Monkey), these pages are concatenated into a single composite image for compatibility.

---

[5]https://github.com/opendatalab/MinerU

We evaluate 19 models across the two configurations. The set includes: **LLMs**: DeepSeek-V3.1 (Deng et al., 2025), GLM-4.5 (GLM et al., 2024), Qwen3 (qwen3-235b-a22b) (Bai et al., 2023), GPT-5 (gpt-5-2025-08-07) (Leon, 2025), Claude-4 (claude-sonnet-4-20250514) (Anthropic, 2025), Gemini-2.5 (gemini-2.5-pro) (Comanici et al., 2025); **Document-Centric Models (DCM)**: DocOwl2(V) (Hu et al., 2024), Docopilot(V) (Duan et al., 2025), Monkey(V) (Li et al., 2024c); **VLMs**: GLM-4.5V (GLM et al., 2024), Qwen3(V), GPT-5(V), Claude-4(V), Gemini-2.5-Pro(V); **RAG Models**: HippoRAG2 (Gutiérrez et al., 2025), MemoRAG (Qian et al., 2025), Raptor (Sarthi et al., 2024), VdocRAG(V) (Tanaka et al., 2025), VisRAG(V) (Yu et al., 2024). Here, "(V)" denotes image-based input.

To maximize performance while leveraging each model's strengths, we impose minimal inference constraints: (1) answers must rely only on the given paper; (2) open-ended responses must be professional, concise, and under 3,000 characters; (3) claim verification outputs must be strictly True or False. Full prompts for two task types are provided in Appendix C.6.

As detailed in Section 3.5, open-ended QA is evaluated with ROUGE-L, BERTScore-F1, Conciseness, Correctness, Completeness, F1-like, and Informativeness, while Claim Verification is measured by accuracy. All results are reported on a standardized 0–100 scale.

## 4.2 Main Results

Table 3 reports all model results on the RPC-Bench test set, highlighting the following findings:

**Traditional surface-matching metrics are insufficient for evaluating paper comprehension, as they fail to capture true semantic understanding.** For example, ROUGE-L and BERTScore cannot reliably distinguish large- from small-scale models (LLMs/VLMs vs. DCM/RAG < 10B). Some models achieve high lexical scores but perform poorly on semantic measures: Monkey(V) attains the best ROUGE-L (20.16%) and strong BERTScore (55.19%), yet its correctness and completeness fall to 17.43% and 11.54%. This shows surface metrics overestimate comprehension, while our framework evaluates semantic correctness and conciseness comprehensively.

**Empirically, LLMs comprehend academic papers better with text-only than with image inputs.** Despite multimodal support, the domain knowledge and intricate structures and layouts in text and images hinder accurate understanding. For example, Qwen3's F1-like score falls from 53.68% (text-only) to 20.28% (image), with conciseness also dropping sharply. GPT-5(V) shows higher conciseness than its text-only version (61.85% vs. 52.68%), but this stems from shorter, less informative outputs caused by reduced correctness and completeness. Overall, the steepest declines appear in correctness and completeness, revealing that current multimodal models still struggle to exploit scholarly visual and textual information.

**Academic paper comprehension is especially challenging for small-scale models (∼8B).** With limited capacity, document-centric models struggle to integrate full-paper content, yielding low F1-like scores (8–18%) and sometimes incoherent outputs—showing that fine-tuning on general-domain data is insufficient for our task. RAG models perform better via targeted retrieval, yet their correctness and completeness remain below 30, indicating limited grasp of essential information.

**Current models struggle to balance correctness, completeness, and conciseness in paper-based QA.** We use Informativeness, a composite metric capturing these dimensions, to assess overall ability. Even top closed-source models score modestly—GPT achieves only 35.05%. Open-source GLM-4.5 shows strong correctness and completeness (57.44%) but low conciseness (37.46%), reducing its Informativeness to 22.41%. These findings highlight significant room for improvement in academic paper comprehension.

## 4.3 Performance across Task Categories

**Performance of Taxonomy-Defined Question Types.** We conduct a fine-grained evaluation of paper comprehension across taxonomy-defined question types. As shown in Figure 3 (a), models perform better on simpler tasks (e.g., concept understanding, method discrimination) than on deeper reasoning tasks, with the gap widening for image-based inputs. Although figures and tables encode rich information, current models struggle to integrate them with long contexts into coherent reasoning. Overall, most models cannot effectively perform contextual reasoning across text and images

Table 3: Evaluation results of free-form QA on the test set. RG-L=ROUGE-L; B-S=BERTScore-F1; Compl. = Completeness; Info. = Informativeness. The best results are highlighted in **bold**, and the second-best results are indicated with underlining.

| Model Type | Model | Traditional | | LLM-as-judge | | | | |
|---|---|---|---|---|---|---|---|---|
| | | R-L | B-S | Conc. | Corr. | Compl. | F1-like | Info. |
| LLM | DeepSeek-V3.1 | 19.12 | **55.98** | 53.15 | 56.10 | 52.17 | 54.06 | 28.73 |
| | GLM-4.5 | 16.03 | 53.18 | 39.02 | 57.55 | 57.32 | 57.44 | 22.41 |
| | Qwen3 | 16.16 | 54.25 | 37.46 | 53.70 | 53.65 | 53.68 | 20.11 |
| | GPT-5 | 16.89 | 54.52 | 52.68 | **67.66** | **65.44** | **66.54** | **35.05** |
| | Claude-4 | 16.60 | 54.02 | 35.78 | 55.74 | 54.98 | 55.36 | 19.81 |
| | Gemini-2.5 | 18.24 | 55.67 | 53.14 | 60.54 | 56.15 | 58.26 | 30.96 |
| DCM | DocOwl2(V) | 14.32 | 46.42 | 50.57 | 11.97 | 6.48 | 8.41 | 4.25 |
| | Docopilot(V) | 16.92 | 53.82 | 38.18 | 18.22 | 17.24 | 17.71 | 6.76 |
| | Monkey(V) | **20.16** | 55.19 | 56.70 | 17.43 | 11.54 | 13.88 | 7.87 |
| VLM | GLM-4.5V | 19.66 | 55.48 | 59.55 | 47.31 | 41.09 | 43.98 | 26.19 |
| | Qwen3(V) | 14.70 | 53.72 | 20.62 | 20.17 | 20.39 | 20.28 | 4.18 |
| | GPT-5(V) | 17.32 | 54.85 | 61.85 | 57.80 | 53.77 | 55.71 | 34.46 |
| | Claude-4(V) | 13.33 | 50.63 | 24.23 | 51.55 | 50.05 | 50.79 | 12.31 |
| | Gemini-2.5(V) | 17.27 | 54.85 | 50.53 | 47.22 | 43.24 | 45.14 | 22.81 |
| RAG | HippoRAG2 | 18.71 | 54.16 | 42.37 | 31.83 | 26.57 | 28.97 | 12.27 |
| | MemoRAG | 13.55 | 52.70 | 49.81 | 23.60 | 18.21 | 20.56 | 10.24 |
| | Raptor | 18.35 | 54.00 | 35.16 | 23.92 | 18.83 | 21.07 | 7.41 |
| | VdocRAG(V) | 17.77 | 52.22 | **65.61** | 21.32 | 13.42 | 16.47 | 10.81 |
| | VisRAG(V) | 16.80 | 54.93 | 36.21 | 24.51 | 21.82 | 23.09 | 8.36 |

to address experimental specific questions (especially experimental analysis), underscoring the need for advances in multimodal paper understanding.

**Results of Claim Verification QA Instances.**    Figure 3 (b) shows the accuracy of baselines on claim verification tasks. Overall, the baseline methods show limited accuracy in claim verification. We find that some multimodal models perform relatively well, likely because they can capture the overall meaning and key claims of a paper. However, large language models may struggle to identify crucial evidence from long contexts. Notably, certain models (e.g., Claude-4, HippoRAG2) exhibit weak instruction-following ability, often failing to output strictly "yes" or "no," which undermines their fact-checking accuracy.

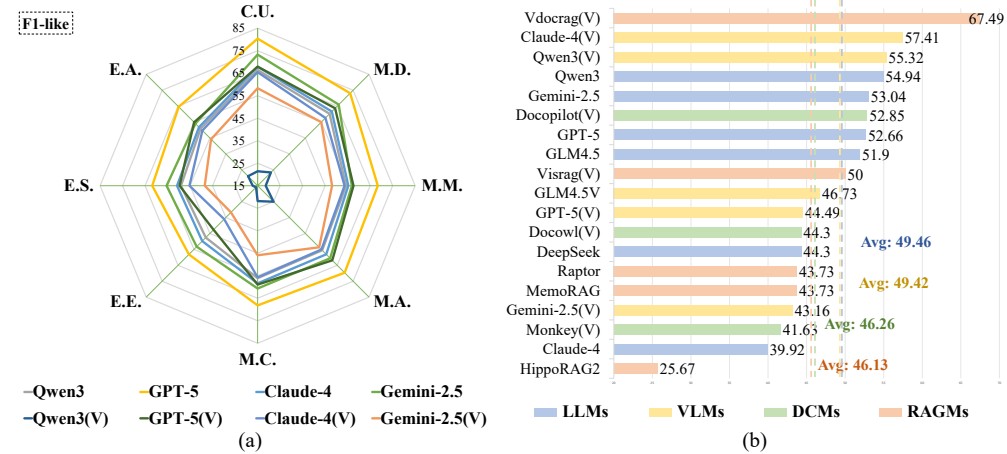

Figure 3: Comparison of LLMs and VLMs on open-ended question answering (F1-like score; left), and the performance of all models on claim verification tasks (ACC; right).

## 4.4 LLM Judgments vs. Human Assessments

To align LLM judgments with human assessments, we sampled 300 open-ended QA instances from the test set and generated predictions from all models in Section 4.1. Although prompts limited answers to 3,000 characters (Appendix B.1), actual output lengths varied widely. To control for length effects, for each QA instance, we selected the three model responses of most similar length and created pairwise comparisons for annotators to judge which answer is better in terms of correctness-completeness. During annotation, both presentation order and left–right placement were randomized, and model identities were masked to reduce bias.

Table 4: Agreement between evaluation configurations and human judgments (AUG: enhancement with title and abstract; SEP/JOI: separate/joint evaluation)

| Setting | P-BT | S-BT | PW-AUC | Avg. |
|---------|------|------|--------|------|
| AUG+SEP | 0.8955 | 0.9137 | 0.7125 | **0.8406** |
| AUG+JOI | 0.9003 | 0.9091 | 0.6966 | 0.8353 |
| RAW+SEP | 0.8773 | 0.9137 | 0.7054 | 0.8321 |
| RAW+JOI | 0.8759 | 0.9137 | 0.7100 | 0.8332 |

Table 5: Analysis of agreement between different LLM judges and human judgments.

| Model | P-BT | S-BT | PW-AUC | Avg. |
|-------|------|------|--------|------|
| GPT-5 | 0.9213 | 0.9137 | 0.7255 | 0.8535 |
| Gemini-2.5 | 0.9164 | 0.9137 | 0.7280 | **0.8527** |
| GLM-4.5 | 0.8971 | 0.9091 | 0.7054 | 0.8372 |

**Consistency Evaluation Metrics.** We measure consistency between model judgments and human assessments using two metrics: **BT-based correlation (P-BT, S-BT)**, which applies the Bradley–Terry model (Turner & Firth, 2012) to convert pairwise outcomes into scores and correlates them with human preferences (Pearson/Spearman); and **pairwise AUC (PW-AUC)**, which directly compares model-predicted pairwise outcomes with human labels.

**Prompt Configuration for LLM Judgments.** We examine prompt design through an ablation of two factors: (1) whether to provide the title and abstract, and (2) whether to present evaluation metrics separately or jointly. Using GLM-4-Plus as the judge, we evaluated four configurations from their cross-combination. As shown in Table 4, including the title and abstract helps the judge understand the question's context, while assessing each dimension independently mitigates error propagation that can arise when an anomalous score affects multiple dimensions in joint evaluation.

**Which LLMs to Judge?** We first used GLM-4-Plus to score 300 sampled QA instances under the configurations above. From these results, we identified the top three models (GPT, Gemini, GLM) and measured their alignment with human assessments (Table 5). The two models with the highest alignment were then jointly chosen as evaluation judges to reduce single-judge bias. This procedure is interpretable, extensible, and adaptable to other tasks as resources allow.

## 4.5 Case Study

We conducted case studies across four key dimensions: (1) common failure modes of current models (Section 4.5.1), (2) detailed analysis for each category (Section 4.5.2), (3) textual versus visual input (Appendix B.4.3), and (4) bottlenecks of RAG methods (Appendix B.4.4)

### 4.5.1 Common Failure Modes

This section illustrates common failure modes of current models in research paper comprehension. Additional examples are provided in Appendix B.4.1.

**Example 1 (Degenerative Output Patterns):** The model's decoding collapses into uninformative content, highlighting the importance of long-form generation tasks to stress-test stability.

**Example 2 (Necessity of Multimodal Grounding):** A text-only model extracts conclusions from text, while a multimodal model grounds claims in visual evidence (e.g., watermarks in Figure 10), revealing cross-modal reasoning capabilities beyond text-only evaluation.

| Example 1 | Example 2 |
|---|---|
| **Paper:** NaturalSpeech 2: Latent Diffusion Models are Natural and Zero-Shot Speech and Singing Synthesizers | **Paper:** AnimateDiff: Animate Your Personalized Text-to-Image Diffusion Models without Specific Tuning |
| **Question:** What are the roles of SoundStream and WaveNet ... andhow do they interact with other components such as the prior model, diffusion model, andaudio codec? | **Question:** What are the visualization results when the domain adapter training (stage 1) is entirely omitted from the process...? |
| **Answer:** SoundStream is used as the neural audio codec ... WaveNet is chosen as the archi-tecture for the diffusion model ... The system consists of three components: a priormodel, a diffusion model, and an audio codec... | **Answer:** The visualizations show that when the domain adapter is entirely removed from the training pipeline, the watermarks from the training dataset appear on the synthetic animations... These results are presented in Appendix B.2 and Fig.10. |
| **VdocRAG(V):** ...The audio codec and the diffusion model interact with the audio decoder to obtain the audio waveform. The audio codec … audio waveform. [The previous sentence is repeated over 100 times, resulting in a 19,867-character response]. | **GPT-5:** Omitting domain adapter training ... This occurs because visual appearance and motion are entangled... (Appendix B.2, Fig. 10, 1st row). **GPT-5(V):** When the domain adapter training (stage 1) is omitted entirely, ... resulting in artifacts such as visible watermarks transferring into the generated visuals (Figure 10). |

Figure 4: Representative case study from the RPC-Bench test set

### 4.5.2 DETAILED ANALYSIS OF EACH CATEGORY

Our evaluation across eight open-ended QA categories reveals clear differences. Detailed examples are provided in Appendix B.4.2.

**Concept Understanding & Method Disambiguation:** Require accurate in-depth understanding. High-capacity LLMs integrate contextual details effectively. VLMs often omit fine-grained elements, DCMs' answers tend to be too general. RAG excels at information extraction but lacks summarization and reasoning capabilities.

**Method Mechanics & Comparison:** LLMs provide clear explanatory contrasts. RAG provides basic comprehension but offers less interpretive depth. VLMs may miss details, DCMs risk repetitive or partial answers.

**Motivation Analysis & Experimental Setup:** Focus on understanding the rationale behind methodological/experimental choices. LLMs and strong VLMs link design choices to their underlying motivation, DCMs and RAG often miss underlying rationale.

**Experiment Exposition & Analysis:** Demand retrieval from dispersed sources, sometimes with quantitative evidence. LLMs can perform qualitative or quantitative analysis of experimental results based on a research paper, VLMs further perform analyses via figure interpretation, RAG ensures detail fidelity. DCMs' answers tend to be too general.

## 5 CONCLUSION

To comprehensively evaluate models' ability to understand research papers, we introduce RPC-Bench, a large-scale benchmark comprising 4,050 papers and 46.3K QA pairs across nine categories, supporting both text and rendered page inputs. We develop an LLM–human collaborative annotation framework to ensure scalability and quality. The evaluation design addresses three key questions—whether to provide titles/abstracts as context, whether to assess metrics independently, and how to select the judging LLM—aiming to better align LLM judgments with human assessments. Our scoring protocol evaluates correctness, completeness, and conciseness. Experiments on 19 state-of-the-art models highlight persistent challenges, including limited use of multimodal information, insufficient conciseness, and weak reasoning over visual content. We envision RPC-Bench as a foundation for evaluating deep understanding and reasoning in large foundation models.

### ETHICS STATEMENT

We confirm that this work adheres to the ICLR Code of Ethics and applicable standards of research integrity. The benchmark introduced in this study (RPC-Bench) was constructed exclusively from publicly available papers and their review–rebuttal pairs hosted on OpenReview. All data were

originally authored for public dissemination, and no private, confidential, or proprietary information was included. No experiments in this work involve human subjects, personal health data, or sensitive demographic attributes.

To ensure fairness, we performed quality control on all collected data, including removal of low-quality or irrelevant content, and balanced sampling to avoid systematic bias in sources. The proposed benchmark does not include any potentially harmful insights, discriminatory content, or security-relevant information. All licenses and usage terms of the source data have been respected, and dataset release will comply with the original terms of access.

Potential conflicts of interest, such as affiliations or sponsorships, have been disclosed in accordance with conference policies. This study aims to advance model evaluation methodology for academic paper comprehension without promoting or enabling malicious applications.

## REPRODUCIBILITY STATEMENT

We have made every effort to ensure that our work is fully reproducible. Full details of the dataset construction process, taxonomy design, annotation procedure, quality control steps, and evaluation framework are described in Section 3 of the main text and further expanded in the Appendix. An anonymous repository containing the PRC dataset (with train/validation/test splits), complete annotation and filtering scripts, and evaluation code for both binary verification and open-ended QA tasks is provided in the main text. Hyperparameter settings and experimental configurations for all evaluated models are also included.

All datasets and source code will be made publicly available upon publication under an open-access license. These resources allow researchers to reproduce the dataset construction, annotation pipeline, and evaluation results reported in this work.

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

## A  USE OF LARGE LANGUAGE MODELS (LLMs)

In this work, large language models (LLMs) were used solely as general-purpose assistive tools for language refinement. Specifically, we employed LLMs to polish the phrasing and improve grammatical correctness in the manuscript. No part of the research design, idea generation, data analysis, experimental execution, or substantive technical writing was performed by LLMs. All conceptual contributions, scientific content, and experiments were conceived, implemented, and verified entirely by the authors.

# B Supplementary Experiments

## B.1 Response Length Analysis

To better understand output length characteristics, we analyzed the distribution of response lengths for all models, as summarized in Table 6. However, most models tended to produce responses approaching the upper bound, which we attribute to their limited ability to comprehend and reason over the research paper content, leading to verbose rather than concise answers. Notably, some models (such as DocOwl2, VdocRAG, and VisRAG) exhibited abnormally high maximum output lengths. Manual inspection revealed that these overly long outputs often consisted of repetitive, non-informative text generated when the model failed to answer the question effectively. Conversely, certain models registered a minimum response length of zero, indicating empty answers either due to refusal triggered by safety policies (e.g., GLM-4.5V) or an inability to provide a response. Overall, the most models struggle to effectively achieve content comprehension, information compression, and logical reasoning within the task constraints, revealing a fundamental gap between the demands of accurate, concise, and contextually grounded scholarly reasoning and the current capabilities of state-of-the-art systems.

Table 6: Response Length Analysis.

| Model Type | Model | Length (char) | | |
|---|---|---|---|---|
| | | Avg | Max. | Min |
| LLM | DeepSeek-V3.1 | 1494.19 | 4350 | 51 |
| | GLM-4.5 | 2127.67 | 10285 | 478 |
| | Qwen3 | 1971.16 | 5527 | 289 |
| | GPT-5 | 1717.14 | 4065 | 37 |
| | Claude-4 | 2109.62 | 5198 | 85 |
| | Gemini-2.5 | 1741.91 | 3985 | 330 |
| DCM | DocOwl2 | 525.52 | 23251 | 0 |
| | Docopilot | 1108.01 | 4031 | 59 |
| | Monkey | 425.10 | 7351 | 10 |
| VLM | GLM-4.5V | 1090.89 | 4079 | 0 |
| | Qwen3(V) | 2019.61 | 7627 | 249 |
| | GPT-5(V) | 1332.63 | 4698 | 24 |
| | Claude-4(V) | 3530.55 | 9853 | 821 |
| | Gemini-2.5(V) | 1667.97 | 3256 | 274 |
| RAG | HippoRAG2 | 1109.48 | 3011 | 39 |
| | MemoRAG | 594.26 | 1957 | 1 |
| | Raptor | 838.01 | 1956 | 0 |
| | VdocRAG | 2391.54 | 30179 | 1 |
| | VisRAG | 1232.17 | 13046 | 2 |
| - | Ground True | 829.63 | 5194 | 5 |

## B.2 Finetune LLM Analysis

We fine-tuned Qwen and LLaMA on the PRC training set, with results summarized in Table 7. Both models achieved consistent improvements in the overall Info. metric, increasing by 11.38% and 10.64%, respectively. Notably, conciseness improved significantly, whereas the F1-like score remained relatively stable. This suggests that, compared to correctness and completeness, models more readily learn to produce concise responses. In contrast, achieving high correctness and completeness imposes greater demands on the models' fundamental comprehension and reasoning capabilities.

Table 7: Performance Comparison in Fine-Tuning Experiments.

| Model Type | Conc. | Corr. | Compl. | F1-like | Info. |
|---|---|---|---|---|---|
| Llama-3.1-8B-Instruct | 41.56 | 34.75 | 30.92 | 32.72 | 13.60 |
| Llama-3.1-8B-Instruct-FT | 77.07 | 36.20 | 29.34 | 32.41 | 24.98 |
| Qwen3-8B | 48.20 | 38.53 | 32.52 | 35.27 | 17.00 |
| Qwen3-8B-FT | 78.58 | 39.30 | 31.82 | 35.17 | 27.64 |

## B.3 EVALUATION OF MODEL CONCISENESS ACROSS TAXONOMY-DEFINED QUESTION TYPES

Figure 5 illustrates the performance of different models across various question types. Most models show minimal variation in conciseness scores across question categories, forming an almost concentric pattern in the radar chart, with no score exceeding 65%. This underscores the difficulty models face in generating responses that are both relevant and precise. Across all question categories, text-based inputs generally yield more concise outputs than image-based inputs. We attribute this to the models' weaker capability in interpreting visual inputs, where the relative loss of explicit textual detail may result in responses that convey less relevant information while being unnecessarily verbose. This pattern is evident in the performance gap between text- and image-based inputs for Qwen3, Gemini-2.5, and Claude-4. An exception is GPT-5(V), which achieves the highest answer conciseness, even surpassing its text-based variant. This result suggests that GPT-5(V) can more effectively leverage visual information, consistent with its strong F1-Like score (see Figure 3 (left)).

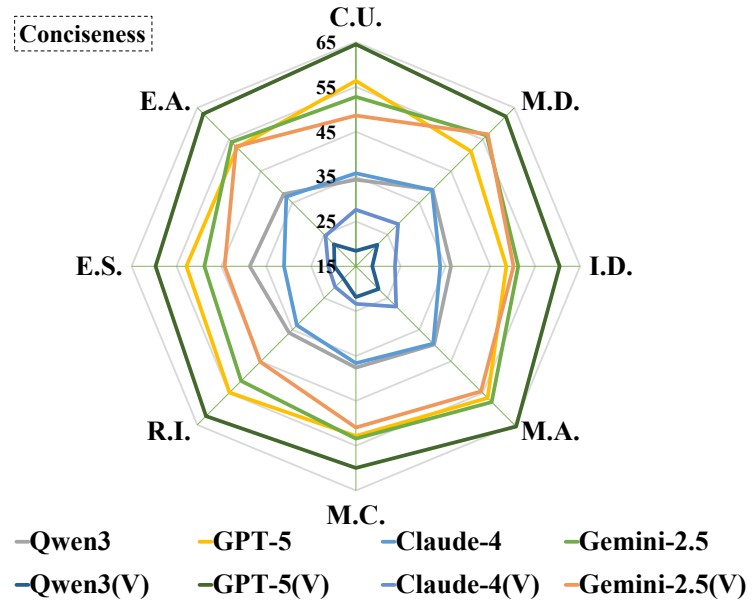

Figure 5: Model Conciseness across Taxonomy-Defined Question Types

## B.4 MORE CASE STUDY

### B.4.1 COMMON FAILURE MODES

Below are two additional case studies:

**Example 3 (Hallucination):** The model sometimes wrongly denies information that is actually present in the source document, highlighting the need for tasks that test precise data extraction and catch such factual-verification errors.

**Example 4 (Precise Output Failures):** Despite the prompt explicitly constraining the output format to strict booleans (True/False), both models violate this requirement: one returns a self-contradictory invalid answer, while the other appends extraneous characters.

| Example 3 | Example 4 |
|---|---|
| **Paper:** Safe RLHF: Safe Reinforcement Learning from Human Feedback | **Paper:** Eureka: Human-Level Reward Design via Coding Large Language Models |
| **Question:** What were the experimental results comparing the paper's method to additional safety RLHF baselines such as Constitutional AI and Safety SFT? | **Question:** Does the EUREKA generate reward functions statistically significantly better than human-designed rewards, based on a Mann-Whitney U-Test with a p-value threshold of 0.05? |
| **Answer:** Win rates for three different methods rated by GPT-4: ... — Constitutional AI — Safety SFT — — Helpfulness Win Rate — 71.8% — 40.2% — 53.6% — ... the experiments revealinteresting insights... | **Answer:** True |
| **Claude-4:** ...the experimental comparison of Safe RLHF to Constitutional AI and Safety SFT baselines was very limited. The paper did not conduct comprehensive experiments comparing their method to Constitutional AI and Safety SFT. | **Claude-4:** FalseTrue

**Claude-4(V):** True1 |

Figure 6: Additional Case study

### B.4.2 DETAILED ANALYSIS OF EACH CATEGORY

Representative examples for each category are presented below.

---

**Example 1: Concept Understanding**

**Question:** What is "cross-LoRA attention" as defined in the paper?

**Answer:** Cross-LoRA attention is an attention mechanism that allows for combining concept and style LoRAs. It splits queries, keys, and values, such that queries from the concept branch attend to keys and values from both the concept and style branches, effectively merging their representations.

**GPT:** Cross-LoRA attention is an attention mechanism designed to combine the concept and style LoRAs... It involves splitting the queries, keys, and values (Q, K, V) into concept and style components. The concept queries attend to the concatenated keys and values from both the concept and style branches...

**GPT(V):** Cross-LoRA attention is the mechanism the authors propose to merge the learned representations of a concept LoRA and a style LoRA during inference... The key idea is to take the query from the concept branch and let it attend to the keys and values from both the concept and style branches.

**DocOwl2:** Cross-LoRA attention is a novel method that enables the model to focus on different aspects of the input image, allowing for more detailed and accurate image generation.

**VisRAG:** Cross-LoRA attention is defined as a mechanism for merging LoRA weights by splitting the query (q) into a concept part (qc) and a style part (qs), and then having the concept query attend to keys and values from both branches.

---

**Example 2: Method Disambiguation**

**Question:** How does the positional embedding used in Rectified Flow Transformers (RFTs) differ from the one used in Diffusion Transformers (DiTs)?

**Answer:** RFTs condition on a continuous time variable 't' from '[0, 1]', while DiTs condition on a discrete noise level 'Σ'. This allows RFTs to leverage the properties of rectified flow for training.

**Claude-4:** The key difference is that RFTs condition on the continuous time variable 't' which ranges from 0 to 1... In contrast, DiTs condition on discrete noise levels 'Σ'.

**Qwen3(V):** Both RFTs and DiTs utilize positional embeddings... However, the paper does not explicitly detail the structural differences... it focuses on the Rectified Flow method itself rather than a component-wise comparison of embeddings.

**DocOwl2:** The positional embedding in RFTs is based on the transformer architecture, while the positional embedding in DiTs is based on the diffusion model architecture.

**HippoRAG2:** Rectified Flow Transformers (RFT) condition on time 't', whereas DiTs condition on the noise level 'Σ'.

## Example 3: Method Mechanics

**Question:** How is normalization applied in the description of Classification Accuracy?

**Answer:** Normalization is performed over the multi-choice options given, rather than over all sequences in the vocabulary.

**Claude-4:** The key aspect of normalization is that it is "performed over the options given, rather than over all sequences." This means that instead of normalizing probabilities across the entire vocabulary... the normalization is restricted to only the specific answer choices provided...

**Claude-4(V):** ...the model uses the probability of the full token sequences corresponding to each class. These probabilities are renormalized to sum to 1 over the provided options for the task... It is done **over the given options only**, rather than over all possible sequences in general...

**Monkey:** Normalization is applied... by dividing the number of correctly classified instances by the total number of instances in the test set.

**MemoRAG:** Normalization is done over the options given, rather than over all sequences.

## Example 4: Motivation Analysis

**Question:** Why was RAFT not tested on tasks such as text-to-SQL or reasoning-based QA, given its potential effectiveness in those domains?

**Answer:** RAFT is specifically designed to enhance the base LLM's performance in domain-specific RAG settings. While it might improve performance in coding or reasoning domains, studying RAFT's performance on those tasks falls outside the scope of this paper, as it focuses on creating domain expert RAG models rather than general post-training strategies.

**Claude-4:** ...RAFT was not tested on text-to-SQL or reasoning-based QA tasks because the authors made specific dataset choices aligned with their research objectives and scope limitations... The paper explicitly states their dataset selection criteria...

**Claude-4(V):** ...due to the focus and scope of this paper, which targets domain-specific open-book question answering (QA) settings... RAFT's method depends on training with **golden and distractor documents**... [This] is fundamentally different from generating executable code (SQL queries)...

**Monkey:** The paper does not provide a direct reason...

**MemoRAG:** RAFT was not tested on... text-to-SQL or reasoning-based QA because the paper's focus is specifically on adapting pre-trained LLMs to domain-specific retrieval-Augmented Generation (RAG) in specialized domains.

## Example 5: Method Comparison

**Question:** How does Self-RAG differ from other RAG methods, such as those that use iterative retrieval or reranking?

**Answer:** Self-RAG differs by adaptively retrieving passages on-the-fly and using self-reflection. Unlike iterative retrieval, it retrieves when needed during generation. Unlike reranking, which happens before generation, Self-RAG's critiques happen post-generation to select the best output. It generates "reflection tokens" to make these decisions.

**Gemini-2.5:** Self-RAG differs by training a single language model to adaptively retrieve passages on an on-demand basis and to **self-reflect on the retrieved passages**... Unlike iterative retrieval which retrieves a fixed number of documents... Self-RAG retrieves passages whenever it deems necessary... Unlike reranking... Self-RAG uses its own critique mechanism after generation...

**Gemini-2.5(V):** Self-RAG differs in that it is an on-demand retrieval system that works during the decoding process... It also generates special tokens ('[Retrieve]', '[Critique]') that allow it to assess its own generation, unlike other methods that re-rank at the start.

**Monkey:** Traditional RAG methods retrieve information, then generate a response. Self-RAG adds a "critique" step where the model evaluates its own generation. However, the paper does not compare it to iterative retrieval.

**HippoRAG2:** Self-RAG is different because it trains one arbitrary LM to generate text and reflect on its own output... by generating special tokens... It retrieves passages one at a time and in an adaptive manner, unlike conventional RAG that retrieves a fixed number of passages.

---

### Example 6: Experimental Exposition

**Question:** What are the effects of not including a weight threshold, and are there specific tasks where this negatively impacts the results?

**Answer:** An ablation study showed minimal impact on most tasks. However, three specific tasks—'Date Understanding,' 'Disambiguation,' and 'Hyperbaton'—showed notable performance declines, with an average decrease of 1.2%.

**GPT-5:** Effect on overall performance: Removing the weight threshold... has minimal impact on most tasks... reduces the average performance across BBH from 34.7 to 33.5... Tasks negatively impacted: Three tasks show notable drops... Date Understanding: $32.9 \rightarrow 17.7$... Disambiguation: $45.2 \rightarrow 40.6$... Hyperbaton: $62.8 \rightarrow 55.5$...

**GPT-5(V):** The paper does not study or report a "weight threshold" mechanism.**... Consequently: - The effects of not using a weight threshold are not evaluated or quantified in the paper. - No tasks are identified where omitting a weight threshold harms performance.

**DocOwl2:** ...including a weight threshold does not significantly affect the method's performance, and there are no specific tasks where this adjustment negatively impacts the results.

**VisRAG:** The removal of the threshold had minimal impact on the majority of tasks... However, three tasks namely Date Understanding, Disambiguation and Hyperbaton exhibited notable effect... average decrease of 1.2%...

---

### Example 7: Experimental Setup

**Question:** Can the success/fitness function be used to initialize the Eureka reward search process?

**Answer:** Yes, the success/fitness function can be used. Section 4.4 describes an experiment where a human - supplied reward function was used to initialize the process.

**Gemini-2.5:** Yes, the success/fitness function can be used to initialize the Eureka reward search process, although it is not the default procedure... Section 4.4 explains that initializing the search with an existing reward function "requires no modification to EUREKA."

**Gemini-2.5(V):** Based on the provided research paper, the success/fitness function isnotused to initialize the Eureka reward search process... The initialization of the reward search process involves providing the Large Language Model (LLM) with two key pieces of information as context: 1. The environment source code. 2. A natural language task description.

**Monkey:** Yes, the success/fitness function can be used to initialize the Eureka reward search process.

**MemoRAG:** Yes, the success/fitness function can be used to initialize the Eureka reward search process.

---

**Example 8: Experimental Analysis**

**Question:** What are the major errors identified in the error analysis conducted in Section 5.1, and how do these errors manifest in the model's predictions?

**Answer:** The major errors identified in the error analysis lie within the click region and scroll direction predictions. The model tends to click the wrong place or scroll in the wrong direction, despite predicting the right action most of the time. This suggests a need for improving the model's ability to understand screen layouts.

**GPT-5:** Section 5.1 identifiesclick region localizationandscroll direction predictionas the major error sources... These errors manifest as the model selecting the correct type of action but failing at the low - level execution...

**GPT-5(V):** Section 5.1 shows that Auto - UI's main errors occur at the execution level, not in deciding what to do. Error types:inaccurate click localizationandincorrect scroll direction... Evidence: averaged category accuracies are 67.4% for click region and 82.0% for scroll direction...

**DocOwl2:** The model's predictions are not accurate because of overfitting and underfitting... identified as model drift and model error...

**MemoRAG:** Major errors include incorrect location prediction during clicks (8%) and misdirection during scroll movements (4%).

### B.4.3 TEXTUAL VS. VISUAL INPUT

This section provides a qualitative comparison of model behavior under purely textual and visual inputs.

Example 1, 2, and 4 shows that LLMs consistently deliver higher factual accuracy, greater answer completeness, and closer alignment with reference answers than VLMs. In Example 3, VLMs accurately identified methodological boundaries (e.g., excluding tasks outside the intended domain-specific RAG scope) and maintained reasoning consistency with the reference. This indicates that VLMs can effectively capture high-level conceptual scope, especially when the task focuses on boundary recognition rather than detailed fact retrieval.

Overall, LLMs excel in tasks requiring precise, evidence-linked, detail-rich answers from textual content. VLMs occasionally match or surpass LLMs in identifying conceptual scope from visual inputs. The observed gap largely reflects VLMs' difficulty in extracting high-precision information from PDF-rendered images of dense academic content (e.g., complex tables, multi-column layouts, long texts), which often results in incomplete or noisy answers compared with clean text inputs.

---

**Example 1**

**Question:** Can the success/fitness function be used to initialize the Eureka reward search process?

**Answer:** Yes, the success/fitness function can be used. Section 4.4 describes an experiment where a human-supplied reward function was used to initialize the process.

**GPT-5:** Yes. EUREKA can be initialized from any provided reward function by substituting it for the first iteration's reward, without modifying the algorithm. The paper demonstrates this "initialization" mechanism with human-written rewards: "we can simply substitute the raw human reward function as the output of the first EUREKA iteration" (Sec. 4.4; App. D.2). In the benchmarks, the success/fitness function F is provided as executable reward code and is used directly as the "Sparse" reward baseline ("These are identical to the fitness functions F...," Sec. 4.1; App. B). Therefore, F can be used in exactly the same way as an initialization for EUREKA: set the first iteration's reward to F, evaluate it, and then let EUREKA's evolutionary search and reward reflection refine it in subsequent iterations (Alg. 1; Sec. 3.2–3.3).

**GPT-5(V):** Yes, the success/fitness function ( F ) can be used to initialize the Eureka reward search process, as outlined in the paper. Eureka explicitly uses ( F ) to evaluate the quality of candidate reward functions generated during its iterative optimization process (Section 3.2). ( F ) measures how effectively a policy derived from a reward function ( R ) solves a task (Definition 2.1). In the

initial phase, the fitness function is employed to assess the performance of reward functions sampled zero-shot from the coding LLM, enabling an informed starting point for the evolutionary search process (Section 3.1). While the initial rewards may exhibit suboptimality relative to the fitness metric, this evaluation ensures that the process begins with executable and task-relevant reward functions, which are progressively refined in subsequent iterations using evolutionary search and reward reflection. Thus, ( F ) plays a critical foundational role in initializing and guiding Eureka's reward optimization.

## Example 2

**Question:** What are the details of LoRA fine-tuning, including the rank, learning rate, and model architecture used?

**Answer:** The details of LoRA fine-tuning include incorporating trainable linear low-rank modules into the query and value projections of each self-attention layer, following Hu et al. (2021). The rank of LoRA is set to 32, the learning rate is $1e - 4$, and the model architecture is the Gecko PaLM model.

**GPT-5:** LoRA fine-tuning details (as used in SQL-PaLM): - Placement: Trainable low-rank adapters inserted into the query and value projections of each self-attention layer. - Rank: 32. - Learning rate: $1e - 4$. - Model architecture: PaLM-2 Gecko (Gecko PaLM model).

**GPT-5(V):** The LoRA (Low-Rank Adaptation) fine-tuning in the SQL-PaLM framework employs the following attributes: 1. Rank: LoRA fine-tuning utilizes a rank of 64, representing the dimensionality of the low-rank decomposition matrices used to adapt large model weights. 2. Learning Rate: A fixed learning rate of 5e-4 is used for the training process with LoRA. 3. Model Architecture: LoRA is applied to the PaLM-2-Large model, a pretrained large-scale Transformer-based architecture. These details align with the authors' exploration of parameter-efficient fine-tuning techniques tailored for scaling Text-to-SQL task performance.

## Example 3

**Question:** Why was RAFT not tested on tasks such as text-to-SQL or reasoning-based QA, given its potential effectiveness in those domains?

**Answer:** RAFT is specifically designed to enhance the base LLM's performance in domain-specific RAG settings. While it might improve performance in coding or reasoning domains, studying RAFT's performance on those tasks falls outside the scope of this paper, as it focuses on creating domain expert RAG models rather than general post-training strategies.

**GPT-5:** Two points from the paper address this: - Reasoning-based QA: RAFT was evaluated on a reasoning-intensive QA benchmark. HotpotQA—an explainable, multi-hop QA dataset—is one of the core evaluations, and RAFT shows large gains there (e.g., 35.28 vs. much lower baselines in Table 1; further analysis in Sections 4.1–4.3). Thus, RAFT was tested on reasoning-based QA within the paper's scope. - Text-to-SQL: The paper's stated focus is domain-specific, document-centric RAG: "we only care about testing the LLM on the same set of documents," and "the LLM will be used to respond to the questions, whose answers can be found within a collection of documents." All methods, data construction, and evaluations are built around reading retrieved documents, citing verbatim spans, and being robust to distractor documents. The chosen datasets (Wikipedia-based QA, API documentation, PubMed) reflect this scope. Tasks like text-to-SQL are not part of this document-retrieval setting the paper studies, so they were outside the evaluation slate.

**GPT-5(V):** Because the paper's goal and problem setting are narrowly defined as domain-specific open-book QA with retrieval. The method and experiments are built around: - Training and testing with a question plus a set of retrieved documents, where "answers can be found within a collection of documents" in the target domain (Domain-Specific Open-Book Exam section). - RAFT's training format requires a golden document D* that contains the answer and supports chain-of-thought with verbatim citations from the relevant document (Section 3). - The evaluation is designed to study reading from documents and robustness to distractors in RAG, using datasets such as PubMed, HotPotQA, and Gorilla APIBench that match this setting (Section 4). Tasks like text-to-SQL or pure reasoning-based QA do not align with this document-grounded RAG setup and do not provide the

golden supporting document and citations RAFT relies on. Hence, they were outside the scope of the study and were not evaluated.

---

### Example 4

**Question:** How does WizardLM address instructions that are not included in the evolved set?

**Answer:** WizardLM addresses instructions not included in the evolved set through In-Breadth Evolving, which evolves entirely new instructions to enhance topic coverage, skill coverage, and overall diversity. This approach is demonstrated in Section 4.5 and Appendix J, showing improved diversity in the evolved instruction data compared to the original data.

**GPT-5:** WizardLM handles missing or failed evolutions in two ways: - During evolution: If an instruction fails to produce a valid evolved version (per the four failure checks), the evolved sample is discarded and the original instruction is put back into the pool "as is," to be attempted again in subsequent evolution epochs. - For training data assembly: After all rounds, the final fine-tuning set is the merge of the initial instruction dataset and all successfully evolved instructions. Thus, instructions that never appear in the evolved set are still included via their original form. Additionally, WizardLM uses In-breadth Evolving to create brand-new, rarer instructions from existing ones, expanding topic and skill coverage to mitigate gaps where specific instructions are not represented in the evolved set.

**GPT-5(V):** WizardLM does not discard instructions that are not part of the evolved set. - During evolution, if an instruction fails to evolve (e.g., gives no information gain, is hard to answer, yields trivial output, or copies words from the prompt), it is "placed back as it is," to be attempted again in later evolution rounds. - When building the final fine-tuning data, the authors merge the initial instruction dataset with all evolved instructions and shuffle them, ensuring unevolved original instructions are still included in training. Thus, instructions not included in the evolved set are retained and reused, both for further evolution attempts and by inclusion in the final fine-tuning corpus.

---

### B.4.4 BOTTLENECKS OF RAG METHODS

We conducted a further analysis of the RAG-based approach, focusing on representative cases:

Example 1: The model successfully retrieved information related to the question but failed to use it effectively in answer generation. Key links between the paper's context and the question were overlooked, and model-generated hallucinations were introduced.

Example 2: The model failed to retrieve the key information needed to answer the question, such as the main innovations of the method. Without this critical material, the generated answer was incomplete or incorrect.

Example 3: The model broke the question down into sub-questions, retrieved all necessary content, and produced a coherent, complete answer.

Overall, The main bottleneck of current RAG-based approach lies in their limited ability to accurately understand complex questions, retrieve targeted knowledge, and integrate information from multiple sources into a unified, meaningful answer.

---

### Example 1

**Question:** Under what conditions does the statement $\mathcal{G}_g \geq 0$ hold, and what could lead to $\mathcal{G}_g < 0$?

**Answer:** The statement $\mathcal{G}_g \geq 0$ holds under the assumption that training and test data come from the same distribution. If this assumption is violated and the test data is simpler (i.e., from a distribution with lower entropy) than the training data, it can lead to $\mathcal{G}_g < 0$.

**Retrieve:** and the gap $\mathcal{G}_r$ is a non-negative value
Note that $p_{\text{aug}}(\boldsymbol{x}') = \mathbb{E}_{\boldsymbol{x} \sim \mathcal{D}_{\text{train}}}[p_{\text{aug}}(\boldsymbol{x}' \mid \boldsymbol{x})]$ only extrapolates from individual data points $x \sim \mathcal{D}_{\text{train}}$ and has density outside the support of $p_{\text{data}}(\boldsymbol{x})$ (e
$\mathcal{G}_a = \mathbb{E}_{\mathbf{x} \sim \mathcal{D}_{\text{test}}}[\text{ELBO}_\theta^*(\mathbf{x})] - \mathbb{E}_{\mathbf{x} \sim \mathcal{D}_{\text{test}}}[\text{ELBO}_\Theta(\mathbf{x})]$
the ELBO on $\mathcal{D}_{\text{train}}$ is greater than or equal to the ELBO on $\mathcal{D}_{\text{test}}$
Therefore the ELBO on $\mathcal{D}_{\text{test}}$ can be higher than the ELBO on $\mathcal{D}_{\text{train}}$ if $\mathcal{D}_{\text{train}}$ and $\mathcal{D}_{\text{test}}$ are not drawn from the same distribution

---

and $\mathcal{D}_{\mathrm{test}}$ has a lower entropy than $\mathcal{D}_{\mathrm{train}}$

Hence, the continuity we gain by replacing $\mathcal{D}_{\mathrm{train}}$ with $p_{\mathrm{DM}}(\boldsymbol{x}')$ is not for free, we lose a small amount of information about $\mathcal{D}_{\mathrm{train}}$

$$\mathcal{G}_{\mathrm{r}} = \mathbb{E}_{\mathbf{x}^{\mathrm{r}} \sim \mathcal{D}_{\mathrm{test}}} \, \mathbb{E}_{\mathbf{x}^{\mathrm{a}} \sim p(\mathbf{x}^{\mathrm{a}}|\mathbf{x}^{\mathrm{r}})} \left[ \mathrm{MS-SSIM} \left( \mathbf{x}^{\mathrm{r}}, \mathbf{x}^{\mathrm{a}} \right) \right]$$

problem (ii) corresponds to density of $\bar{p}_{\mathrm{aug}}(\boldsymbol{x}')$ outside of $p_{\mathrm{data}}(\boldsymbol{x})$

For VAEs, we refer to the difference between training and test set ELBO as the generalization gap

Therefore, $\mathcal{G}_{\mathrm{g}} \geq 0$ A smaller $\mathcal{G}_{\mathrm{g}}$ corresponds to a better generalization performance of a VAE Remark (Test data entropy can also affect the ELBO value)

**Output:** ... ( $\mathcal{G}_{\mathrm{g}} \geq 0$ ) holds when player ( i ) maintains sufficient SCs and units, avoiding disbandment; ( $\mathcal{G}_{\mathrm{g}} < 0$ ) occurs if player ( i ) has few SCs, disbands units, or is effectively blocked by player ( -i ).

## Example 2

**Question:** What are the novel aspects of the proposed method compared to prior work such as Lee et al. (2019), and how do they contribute to the improvement in performance?

**Answer:** ... The improvement in performance over ICT mainly comes from three factors: using MoCo to handle negatives, which allows scaling to a large number of negatives; the sampling procedure to generate pairs of (query, key); and using data from both CC-net and Wikipedia for training. The paper will be updated to highlight the origin of these gains. ...

**Retrieve:** A downside of this approach is that it requires extremely large batch sizes to work well Chen et al (2020), with Qu et al (2021) reporting improvement in the context of information retrieval up to 8192 negatives This method has been widely used to train information retrieval models with supervised data Chen et al (2017b); Karpukhin et al (2020) and was also considered when using ICT to pre-train retrievers by Lee et al

Hence, we believe that most of the improvement compared to the state-of-the-art retrievers can be attributed to our contrastive pre-training strategy Table 7: Impact of data augmentions We report nDCG@10 without fine-tuning on MS MARCO

Third, when used as a pre-training method before fine-tuning on MS MARCO, our technique leads to strong performance on the BEIR benchmark We perform ablations to motivate our design choices, and show that cropping works better than the inverse Cloze task Finally we train a multilingual dense retriever with contrastive learning and show that it achieves state-of-the-art performance Code and pre-trained models are available here: https://github com/facebookresearch/contriever 2 Related work

It should be noted that our fine-tuning procedure on MS MARCO is simpler than for other retrievers, as we use a simple strategy for negative mining and do not use distillation Our model would probably also benefits from improvements proposed by these retrievers, but this is beyond the scope of this paper Finally, we illustrate the benefit of our retriever compared to BM25 in a few-shot setting, where we have access to a small number of in-domain retrieval examples

Interestingly, as shown in Table 7, the random cropping strategy outperforms the inverse cloze task in our setting We believe that random cropping, leading to the identical distributions of keys and queries, leads to more stable training with MoCo compared to ICT This might explains part of the difference of performance between the two methods We also investigate whether additional data perturbations, such as random word deletion or replacement, are beneficial for retrieval Training data

In this section, we briefly review relevant work in information retrieval, and application of machine learning to this problem This is not an exhaustive review, and we refer the reader to Manning et al (2008), Mitra et al (2018) and Lin et al (2020) for a more complete introduction to the field Term-frequency based information retrieval

The methods described in the previous paragraph were applied to re-rank documents, which were retrieved with a traditional IR system such as BM25 Gillick et al (2018) first studied whether continuous retrievers, based on bi-encoder neural models, could be viable alternative to re-ranking In the context of question answering, Karpukhin et al (2020) introduced a dense passage retriever (DPR) based on the bi-encoder architecture

We report results, without fine-tuning on MS MARCO in Table 6 We observe that the difference of performance between the two methods is small, especially after fine-tuning on MS MARCO We thus propose to use MoCo as our contrastive learning framework, since it scales to a larger number of negative examples without the need to increase the batch size Number of negative examples

Finally, we also consider additional data augmentations such as random word deletion, replacement or masking We use these perturbations in addition to random cropping 3 1 3 Building large set of negative

pairs An important aspect of contrastive learning is to sample a large set of negatives Most standard frameworks differ from each other in terms of how the negatives are handled, and we briefly describe two of them, in-batch negative sampling and MoCo, that we use in this work

arXiv preprint arXiv:2002 08909, 2020 3, 5, 7 Kaiming He, Haoqi Fan, Yuxin Wu, Saining Xie, and Ross Girshick Momentum contrast for unsupervised visual representation learning In Proceedings of the IEEE/CVF Conference on Computer Vision and Pattern Recognition, pp 9729–9738, 2020

**Output:** The proposed method introduces a DRL-based heuristic that generates local moves via a deep policy network, learns longer-sighted search policies, utilizes a GNN-based representation for JSSP, and employs a message-passing mechanism for efficient solution evaluation, leading to improved performance over prior methods.

## Example 3

**Question:** What is the purpose and methodology of the In-breadth Evolving analysis, and how does it demonstrate the diversity of the dataset compared to ShareGPT and Alpaca (Self-Instruct)?

**Answer:** The Analysis of In-breadth Evolving section aims to enhance topic coverage, skill coverage, and overall dataset diversity. The methodology involves using BERT to encode instructions into 768-dimensional embeddings, applying t-SNE to reduce the embeddings to 2 dimensions, and using k-means clustering to partition instructions into 20 clusters for visualization. As shown in Figure 7, the data points of the dataset are more dispersed than those of ShareGPT and Alpaca (Self-Instruct), indicating better topic diversity.

**Retrieve:** In-breadth Evolving aims to enhance topic coverage, skill coverage, and overall dataset diversity

Here is instruction of Example N  Rewritten Prompt: In-Breadth Evolving aims to enhance topic coverage, skill coverage, and overall dataset diversity Open-domain instruction finetune datasets (e g , Alpaca, ShareGPT, etc ) are typically small in scale, lacking topic and skill diversity To solve this problem, we designed a prompt to generate a completely new instruction based on the given instruction, requiring the new instruction to be more long-tailed

As shown in the Figure 7, the data points of our dataset are more dispersed than ShareGPT and Alpaca (Self-Instruct), which indicates the better topic diversity in our instructions [](images/3c88da3581f2a7f8ccd6105127fb98eef3bcb4ba3c4205b76aaad6ceef6e492a jpg)

The results indicate that (i) the ShareGPT is a better seed for evol-instruct than Alpaca, (ii) larger evolved data size can improve model capacity, and (iii) our proposed Evol-Instruct method is not dependent on ChatGPT, other strong open source model such as Llama-2 is also a good substitute for ¡table¿¡tr¿¡td¿Model¡/td¿¡td¿Avg

, Llama-1 65B, Llama-2, Mistral-7B) indicate that our Evol-Instruct can be widely applied to various pre-trained models Analysis of In-depth Evolving The Figure 5a and 5b presents an ablation study investigating the impact of the number of data evolution rounds To study the depth of the evolving process, we use ChatGPT to judge the difficulty level of instruction The used prompt please refer to Appendix E

, 2023a), and Tulu (Wang et al , 2023) 4 2 EXPERIMENT DETAIL To construct the dataset, we initialize it with the $52k$ instruction dataset of Alpaca and iteratively perform $M$ evolutions, where $M = 4$ For each instruction in each round of evolution, we randomly select one evolving prompt from total six prompts (i e

Table 2: WizardLM with different data seed, data size, evol model, and base model size [](images/0e2a6d6f9e7ed060e716f7bc3d0b1fc4001af54061952a7dcfec6a8a4f09a6e1 jpg) Figure 5: The difficulty level between ShareGPT, Alpaca, and our four epochs of evolved instruction

In order to study the impact of different data seeds Evol models scale of evolved dataset pre-trained models on our proposed method we conducted the following experiments a) Using 70k ShareGPT as the seed data to obtain WizardLM-13b (ShareGPT Seed) b) Using LlaMA-2-70B-Chat to replace ChatGPT as the evolutionary execution model to obtain WizardLM-13b (LlaMA-2-70B-Chat Evol)

To examine (qualitative analysis) the breadth (diversity) of different dataset, we firstly use BERT to encode each instruction and get its embedding with 768 dimensions, then use a dimension reduction algorithm named t-SNE to reduce embedding dimension to 2, finally we apply a clustering algorithm **k** -means to partition the instructions of each dataset into 20 clusters for an intuitive visualization

Alpaca used a dataset of 50k instructions generated from a limited (e g , 175 samples) seed set of manually-written instructions Our work is different from InstructGPT and Vicuna in that we use AI-generated data for instruction fine-tuning Unlike Alpaca's self-instruct Wang et al (2022a) generation method, Evol-Instruct can control the difficulty and complexity level of the generated instructions

**Output:** The purpose of the In-breadth Evolving analysis is to enhance topic coverage, skill coverage, and overall dataset diversity. The methodology involves encoding each instruction using BERT to obtain 768-dimensional embeddings, reducing the dimensions to 2 using the t-SNE algorithm, and then applying the $k$-means clustering algorithm to partition the instructions into 20 clusters for visualization. The analysis demonstrates the diversity of the dataset by showing that the data points in the cluster scatter plot are more dispersed compared to ShareGPT and Alpaca (Self-Instruct), indicating better topic diversity in the instructions.

## C    BENCHMARK DETAILS

### C.1    QUALITY-BASED FILTERING CRITERIA

We perform quality-based filtering at two distinct stages (comment–response pairs and QA items) to remove low-quality items that cannot be reliably answered using only the paper's content. The filtering criteria are:

- Temporary or editorial issue: corrections of grammar/spelling errors (e.g., "We corrected 'benchamrks' to 'benchmarks'"), figure color/font adjustments, formatting changes or adding references, open-sourcing code/data (e.g., "Added reference to Smith et al."), where the response merely acknowledges the fix without academic substance.

- External resource dependency: responses whose validity depends on external materials not contained in the paper (e.g., "More cases: https://..."), or indirect or evasive replies (e.g., "See Section X").

- Non-substantive commitments: promises of future additions (e.g., "We will add a limitations section", "Will address in future work") without providing specific details or a concrete resolution in the current submission.

### C.2    ANNOTATION PLATFORM

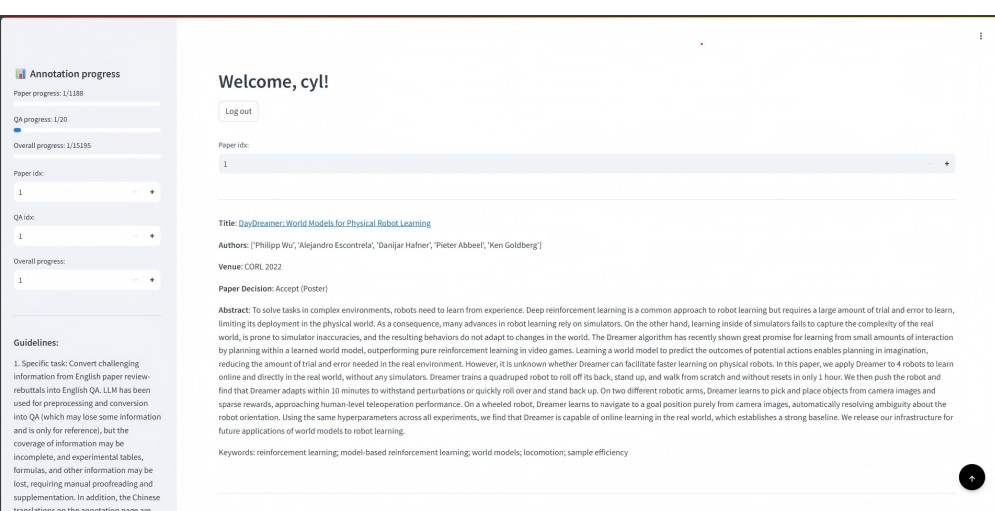

Figure 7: Screenshot of the Annotation Interface 1

### C.3    REVIEW PLATFORM

### C.4    DECOMPOSE PROMPT

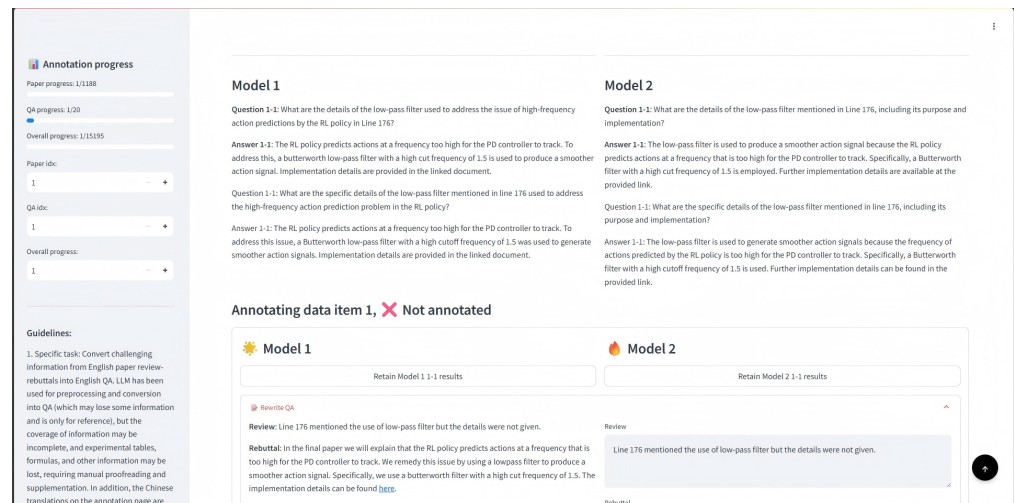

Figure 8: Screenshot of the Annotation Interface 2

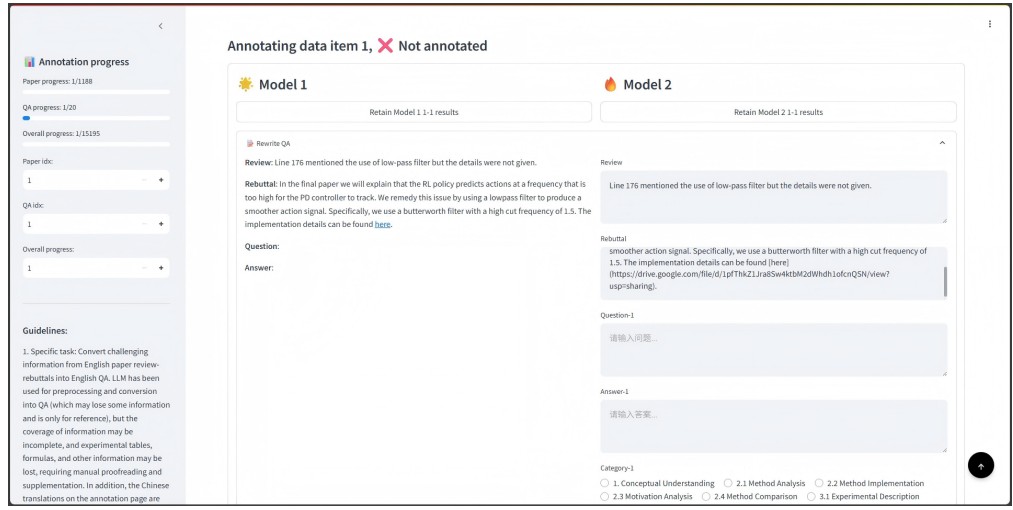

Figure 9: Screenshot of the Annotation Interface 3

```
You are an excellent reviewer of papers. You are tasked with extracting
    QA pairs from the "review", "rebuttal" and "extra_rebuttal" sections
    of a conference paper submission. This process includes identifying "
    review" provided by reviewers and pairing them with the corresponding
     answers authored by the paper's authors, utilizing content from both
     the "rebuttal" and any relevant "extra_rebuttal" sections.
Your goals are: Extract and classify the QA pairs. Ensure that references
     and citations in the rebuttal are preserved in their original format
     within the answers, maintaining the academic rigor and clarity.
    Determine whether each question-answer pair is 'multimodal-related,'
    a broad concept that includes questions explicitly about the figures
    and tables in the paper or questions that can only be answered by
    referring to the contents of these figures and tables.

Input Structure:

review: Concatenation of all reviews, including multifaceted evaluations
    of the paper and any responses or questions directed at the authors'
    rebuttal.
rebuttal: The content in the rebuttal is a concatenation of the answers
    to all the review questions.
```

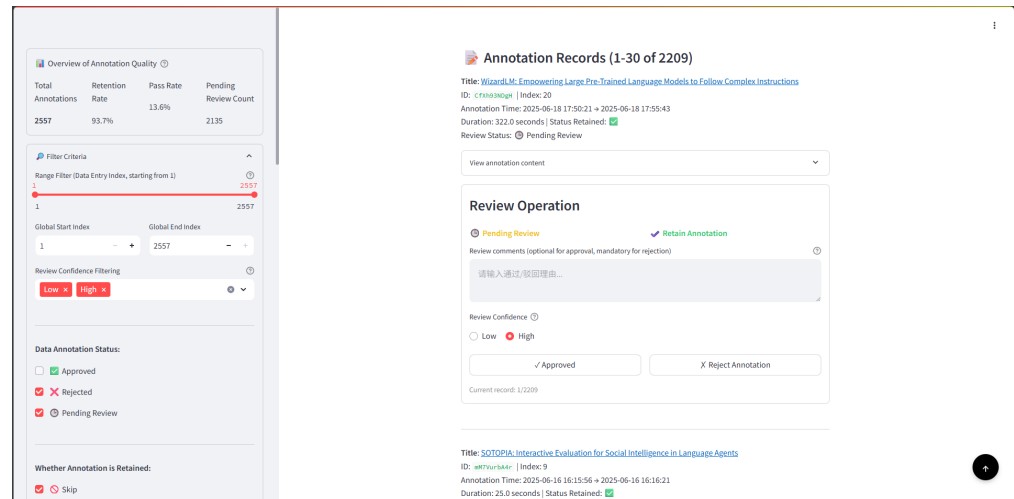

Figure 10: Screenshot of the Review Interface 1

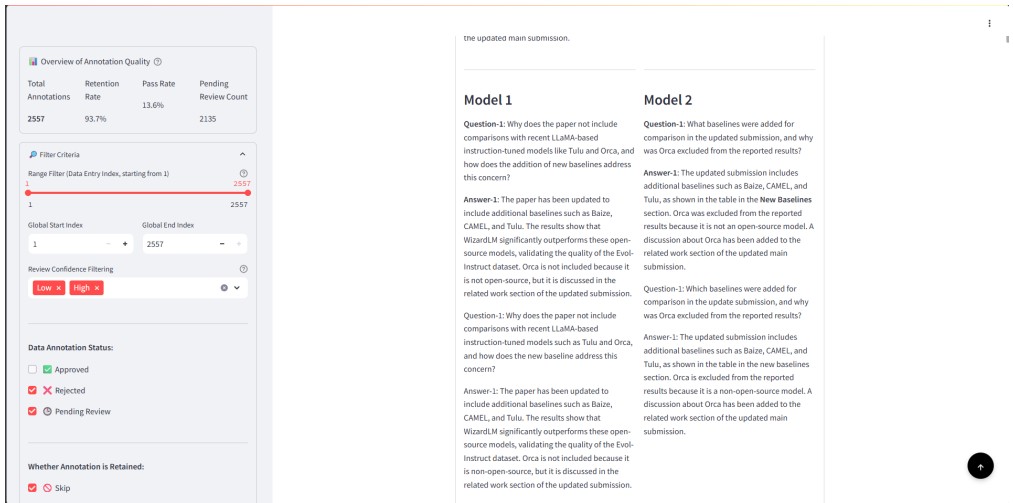

Figure 11: Screenshot of the Review Interface 2

```
extra_rebuttal: Additional content from the authors that may cover the
    current questions.

Output Requirements:

For each QA pair, output in the following JSON format:
[
    {
        "question": "extracted question text here",
        "answer": "corresponding answer text here",
        "is_multimodal_related": true or false
    },
    ...
]

Guidelines:

1. Split combined questions into finer sub-questions for clarity but
    merge them if they cannot stand alone meaningfully.
2. Ensure the completeness and consistency of the extracted QA pairs.
3. Use content from the extra_rebuttal to enhance or clarify answers when
    applicable and relevant to the question.
```

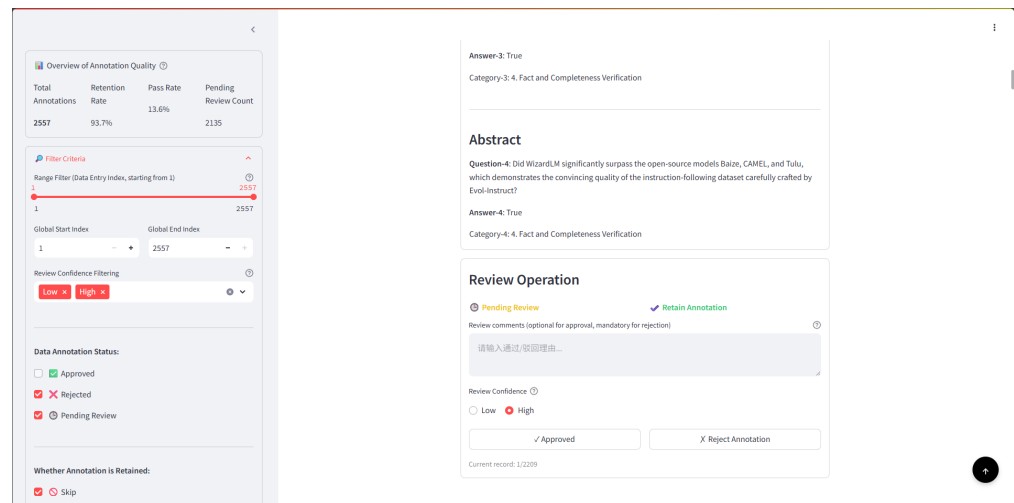

Figure 12: Screenshot of the Review Interface 3

```
4. Ensure that the rebuttal content is fully utilized in the answers,
   forming comprehensive and clear QA pairs that correspond to the
   questions posed.
5. Use your judgment to label each QA pair as 'multimodal-related' if it
   either explicitly poses questions about the figures and tables in the
   paper or implicitly requires the content of these figures and tables
   to answer the question.
6. The answers should be as comprehensive as possible, retaining any
   relevant content such as "references" that can assist in addressing
   the questions.
7. Use the original content from the review, rebuttal, and extra_rebuttal
   to construct the QA pairs, avoiding unnecessary modifications to the
   original text.

Input:
review: It is novel enough to combine the advantages of two famous models
   (Transformer, RNN). Also, the combining method looks applicable to a
   variety of scenarios. The experimental results are impressive,
   showing superior performance to previous Transformer.

I think the draft would become better if there is a more complete
   explanation and figures about the self-attention with recurrence (RSA
   ) operation.

I think the novelty of this draft is enough for the publication and the
   experimental results are impressive. English is good enough as well.
   I recommend weak accept for the draft.

rebuttal: Thanks for your encouraging words and constructive comments. We
   sincerely appreciate your time in reading the paper, and our point-
   to-point responses to your comments are given below.

> I think the draft would become better if there is a more complete
   explanation and figures about the self-attention with recurrence (RSA
   ) operation.

Thank you for this instructive comment. Following your suggestions, we
   have provided a graphical illustration of a single headed RSA module
   in Figure 1 (d) on Page 2, and a more detailed explanation about the
   operation of RSA has been given in the paragraph of "Operation of
   multihead RSA modules" on Page 5.
```

```
In the meanwhile, we have also reorganized the whole Section 3 to better
    explain the proposed RSA. Specifically,
For a single head RSA, we have devoted a paragraph right after equation
    (4) to detail the different types of REMs i.e. $\mathbf{P}$ in the
    paper.

For your easy reference, we have listed the multihead RSA operation below
    :

        Procedure for the Multihead RSA
                    - Choose masked or unmasked REMs according to the nature
                      of the task.
                    - Select the hyperparameters including the dilating
                      factor $d$ and the numbers of the six types of REMs $
                      (k_1,\dots,k_6)$.
                    - For each head, apply equation (4) with a different REM.
                    - Apply a linear layer to combine the output from all
                      heads, and perform layer-normalization and dropout.

extra_rebuttal: We will make the following revisions to the paper:

1. Block-Recurrent Transformer (BRT) [1] has been adopted as another
    baseline model for the NLP experiment in Section 4.3, and its results
    are presented as follows.

|                            | BRT        | RSA-BRT    |
| -------------------------- | ---------- | ---------- |
| Enwik8                     | 1.0746     | **1.0683** |
| Text8                      | 1.1652 | **1.1625** |
| WikiText-103               | 23.758     | **23.639** |
| # Averaged Params added (%) |           | 8.68E-05   |

It can be seen that RSA-BRT exceeds the baseline BRT's performance on all
    datasets.

**The results of this table will be used to fill in the blanks in Table 3
    (b) of the paper.**

2. Two additional experiments for Section 4.4 have been conducted during
    the second discussion phase, which are detailed in the responses to
    Reviewers mvWh and Zrmk.

(1) A scaling experiment is conducted for RSA-BRT v/s BRT on Enwik8
    dataset. The results are shown as follows.

| #  layers          | 8          |           | 10         |           |
    12           |          | 14          |           |
| ------------------ | ---------- | --------- | ---------- | --------- |
    ---------- | --------- | ---------- | --------- |
|                    | Params     | BPC       | Params     | BPC       |
    Params     | BPC       | Params     | BPC       |
| BRT                | 35,080,908 | 1.127     | 41,905,868 | 1.106     |
    48,730,828 | 1.098     | 55,555,788 | 1.079     |
| RSA-BRT            | 35,080,943 | **1.120** | 41,905,913 | **1.104** |
    48,730,883 | **1.092** | 55,555,853 | **1.072** |
| Increase in #Params | 35        |           | 45         |           |
    55           |          | 65          |           |

It can be seen that, with only less than 100 new parameters, RSA-BRT can
    achieve some improvement over the baseline BRT. More importantly, the
    advantage can be consistently observed for all model sizes.
```

(2) Another scaling experiment is conducted for RSA-XL against TL-XL on
    Text8 dataset, where REM is replaced by a learnable Toeplitz matrix
    in the latter model. The results are shown as follows.

| #  layers          | 8          | | 10         | |
    12          | | 14         | |
| ------------------ | ---------- | --------- | ---------- | --------- |
    ---------- | --------- | ---------- | --------- |
|                    | Params     | BPC       | Params     | BPC       |
    Params     | BPC       | Params     | BPC       |
| TL-XL              | 34,180,645 | 1.193     | 41,013,799 | 1.188     |
    47,846,953 | 1.183     | 54,680,107 | 1.178     |
| RSA-XL             | 34,139,725 | **1.181** | 40,964,695 | **1.170** |
    47,789,665 | **1.164** | 54,614,635 | **1.160** |
| Decrease in #Params | 40,920    |           | 49,104     |           |
    57,288     |           | 65,472     |           |

From the above table, it can be seen that the newly added TL-XL also
    performs worse than the RSA-XL of a similar model size, indicating
    parameter redundancy. In other words, RSA-XL enjoys a much better
    parameter-efficiency.

**These two experiments will be further included into Section 4.4 of the
    paper.**

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

    Processing Systems.

Output:
[
    {
        "question": "I think the draft would become better if there is a
            more complete explanation and figures about the self-
            attention with recurrence (RSA) operation.",
        "answer": "Thank you for this instructive comment. Following your
            suggestions, we have provided a graphical illustration of a
            single headed RSA module in Figure 1 (d) on Page 2, and a
            more detailed explanation about the operation of RSA has been
            given in the paragraph of 'Operation of multihead RSA
            modules' on Page 5. In the meanwhile, we have also
            reorganized the whole Section 3 to better explain the
            proposed RSA. Specifically, for a single head RSA, we have
            devoted a paragraph right after equation (4) to detail the
            different types of REMs i.e. $\\mathbf{P}$ in the paper. For
            your easy reference, we have listed the multihead RSA
            operation below: Procedure for the Multihead RSA - Choose
            masked or unmasked REMs according to the nature of the task.
            - Select the hyperparameters including the dilating factor
            $d$ and the numbers of the six types of REMs $(k_1,\\dots,k_6
            )$. - For each head, apply equation (4) with a different REM.
            - Apply a linear layer to combine the output from all heads,
            and perform layer-normalization and dropout.",
        "is_multimodal_related": true
    }
]

Input:
review: I would like to request further clarification regarding your
    paper after carefully reading it. Firstly, I would like to express my

```
        sincere appreciation for the captivating nature of your work and the
        clarity with which it is presented. Congratulations for the
     acceptance of your paper into the top 5% category.

In Section 4.3, I noticed the utilization of Transformer-XL with 14
     layers, resulting in a notable achievement of 1.074 on the Enwik8
     dataset. However, upon referencing the Transformer-XL paper, it
     became apparent that they reported lower bpc values, specifically
     1.06 with 12 layers, 1.03 bpc with 18 layers, and an impressive 0.99
     bpc with 24 layers.

To enhance my understanding, I kindly request your insights regarding the
        decision to opt for 14 layers instead and the possible reasons
     behind the relatively higher bpc despite employing deeper layers.
     Additionally, I would greatly appreciate any additional details or
     insights you can provide to address these inquiries.

Thank you in advance for your time and consideration. Your input will
     greatly contribute to my comprehension of your valuable research.
     Once again, congratulations on the successful publication of your
     paper.
rebuttal: Hi Lokesh, thanks for the question!

The observed difference between the reported bits per character (bpc) for
     Enwik8 in Section 4.3 of our paper and the original Transformer-XL
     paper can be attributed to our decision to utilize Nvidia's
     implemented Transformer-XL (https://catalog.ngc.nvidia.com/orgs/
     nvidia/resources/transformerxl_for_pytorch) rather than the official
     repository. We chose the Nvidia version due to its enhanced user-
     friendliness and comprehensive multi-card support.

However, it is important to note that the reproduction by Nvidia resulted
     in slightly worse bpc for Enwik8 compared to the figures reported in
     the original paper. Specifically, the bpc for Enwik8 with a 12-layer
     Transformer-XL exceeded the previously reported value of 1.06. This
     discrepancy could be attributed to variations in the implementation
     and environment between Nvidia's version and the official repository.

Furthermore, from an intuitive perspective, when a model is
     overparameterized, the proposed RSA may exhibit better generalization
     ability, as illustrated in Figure 1. In order to emphasize the
     benefits of the proposed RSA, we employed a slightly larger model.
     Unfortunately, due to limited resources, we were unable to conduct
     further experiments using a 24-layer XL model.

While acknowledging these limitations, we believe that the use of Nvidia'
     s implementation, combined with our modifications, provides valuable
     insights and supports our argument. The comparison between the
     modified models, despite the slight deviations, offers meaningful
     observations regarding the potential advantages of the proposed RSA.

extra_rebuttal:
Output:
[
    {
        "question": "In Section 4.3, I noticed the utilization of
            Transformer-XL with 14 layers, resulting in a notable
            achievement of 1.074 on the Enwik8 dataset. However, upon
            referencing the Transformer-XL paper, it became apparent that
             they reported lower bpc values, specifically 1.06 with 12
            layers, 1.03 bpc with 18 layers, and an impressive 0.99 bpc
            with 24 layers. To enhance my understanding, I kindly request
             your insights regarding the decision to opt for 14 layers
            instead and the possible reasons behind the relatively higher
             bpc despite employing deeper layers.",
```

```
            "answer": "The observed difference between the reported bits per
                character (bpc) for Enwik8 in Section 4.3 of our paper and
                the original Transformer-XL paper can be attributed to our
                decision to utilize Nvidia's implemented Transformer-XL (
                https://catalog.ngc.nvidia.com/orgs/nvidia/resources/
                transformerxl_for_pytorch) rather than the official
                repository. We chose the Nvidia version due to its enhanced
                user-friendliness and comprehensive multi-card support.
                However, it is important to note that the reproduction by
                Nvidia resulted in slightly worse bpc for Enwik8 compared to
                the figures reported in the original paper. Specifically, the
                 bpc for Enwik8 with a 12-layer Transformer-XL exceeded the
                previously reported value of 1.06. This discrepancy could be
                attributed to variations in the implementation and
                environment between Nvidia's version and the official
                repository. Furthermore, from an intuitive perspective, when
                a model is overparameterized, the proposed RSA may exhibit
                better generalization ability, as illustrated in Figure 1. In
                 order to emphasize the benefits of the proposed RSA, we
                employed a slightly larger model. Unfortunately, due to
                limited resources, we were unable to conduct further
                experiments using a 24-layer XL model. While acknowledging
                these limitations, we believe that the use of Nvidia's
                implementation, combined with our modifications, provides
                valuable insights and supports our argument. The comparison
                between the modified models, despite the slight deviations,
                offers meaningful observations regarding the potential
                advantages of the proposed RSA.",
            "is_multimodal_related": true
        }
]

Input:
review: -        The idea of utilizing dataset exchangeability to identify
        test set contamination is novel and interesting.
-        The proposed sharded likelihood comparison test addresses the
        tradeoff between statistical power and computational requirements of
        the permutation test, which is promising. The sharded rank comparison
         test also provides (asymptotic) guarantees on false positive rates.
-        Experimental results are promising. A GPT-2 model is trained from
         scratch on standard pretraining data and known test sets to verify
        the efficiency of the proposed method in identifying test set
        contamination. The method is also tested with an existing model,
        LLaMA2, on the MMLU dataset, showing general agreement with the
        contamination study results.
-        Although a more efficient sharded rank comparison test is
        proposed, the computational complexity is still considerable. For
        example, testing 49 files using 1000 permutations per shard can take
        12 hours for LLaMA2.
-        There is no comparison with other baseline methods.
-        The method relies on a strong assumption of data exchangeability,
         which may not hold in real-world datasets.
If a dataset is not exchangeable, how effective is the method?

rebuttal: Thank you for your thorough review and valuable feedback on our
        work.

We'd like to address the concern regarding the computational complexity
        of our test. It's important to note that the test is a one-time
        process for any given model and dataset; once the p-values are
        computed, there is no need for recalculation. Our findings indicate
        that a number of permutations beyond 30-50 per shard offers
        diminishing returns, as shown in Figure 3 (right).
```

Furthermore, the test's design allows for easy parallelization. Each
    shard permutation can be evaluated independently, enabling the use of
     inexpensive commodity hardware to run the test significantly faster.

Regarding the assumption of data exchangeability, this is a strictly
    weaker condition than the commonly held assumption of independent and
     identically distributed (I.I.D.) data in machine learning. Most
    datasets satisfy this assumption to some extent.

We acknowledge the validity of our test hinges on data exchangeability.
    However, depending on the source of non-exchangeability, it is often
    the case that a dataset can be altered slightly so that our test is
    still valid. For example, a common source of non-exchangeability is
    the presence of ascending IDs (e.g., as in SQuAD and HumanEval). We
    can adjust the d a t a by either removing these IDs or permuting the
    examples while keeping IDs c o n s t a n t to retain the test's
    applicability. This is discussed in more detail in the revised paper.

Finally, we appreciate your suggestion to include baseline comparisons.
    We provide a comparison against a contamination detection method
    called Min-K% Prob, a state of the art heuristic method for
    contamination detection in language models proposed contemporaneous
    to our work by Shi et. al. (2023).

We find that our method matches or exceeds the performance of this state
    of the art heuristic method. Please see the table in the top-level
    comment for numbers.

extra_rebuttal: We are sincerely grateful to the reviewers for dedicating
     their time and effort to review our work, and we appreciate the
    recognition of the novelty of using exchangeability for contamination
     detection and the significance of our contribution given the
    discourse surrounding contamination in the field. We address each
    reviewer's comments in detail below. We have made numerous updates to
     the submission, most notably with the results of our test on four
    popular open models and eight commonly used benchmarks.

One question shared by multiple reviewers is regarding the exact notion
    of contamination we consider in this work. Rather than consider a
    definition based on heuristics like n-gram overlap, we consider
    contamination detection as the problem of detecting statistical
    dependence between the test data and model parameters. Within this
    setting, our work shows that it is possible to provide provable
    guarantees of contamination in the case of verbatim contamination,
    where the full test set (with examples and labels) is embedded in the
     pretraining data.

To illustrate the relevance of this setting, we note that a search of The
     Pile, a large open-source language modeling dataset, yielded
    numerous instances of small real-world datasets embedded with
    examples appearing in-order. As one example, the following is an
    excerpt from a dataset for an annotation tool made by Explosion, the
    creators of spaCy, a popular natural language processing framework,
    found in The Pile:

```
{"text":"Uber\u2019s Lesson: Silicon Valley\u2019s Start-Up Machine Needs
    Fixing","meta":{"source":"The New York Times"}}
{"text":"Pearl Automation, Founded by Apple Veterans, Shuts Down","meta
    ":{"source":"The New York Times"}}
{"text":"How Silicon Valley Pushed Coding Into American Classrooms","meta
    ":{"source":"The New York Times"}}

Source: https://github.com/explosion/prodigy-recipes/tree/
    fc06f6a6d93bc477e98cf0d8357c39322e4f5a6a
```

```
What our work shows is that by exploiting exchangeability in this setting
    , we are able to provide guarantees on the false positive rate of our
     test.

Multiple reviewers indicated the desire for a comparison against a
    baseline method. While no other existing work is comparable in the
    sense that it provides a statistical proof of contamination like ours
    , we provide a comparison against a state of the art heuristic method
     for contamination detection called Min-K% Prob, proposed by Shi et.
    al. (2023) contemporaneous to our work. We use the same pretrained
    model and test sets from our experiments in Section 4.1.

| Dataset     | Duplication Count | Sharded p (ours) | Percent
    Contaminated (Min-K%-Prob) |
|------------|------------------|-----------------|--------------------------------------|

| BoolQ     | 1                | 0.156           | 3%
                  |
| HellaSwag | 1                | 0.478           | 2%
                  |
| MNLI      | 10               | 1.96e-11         | 100%
                  |
| MMLU-Pro-Law | 50            |  1e-38           | 90%
               |
| MMLU-HS-Psych | 100          |  1e-38           | 74% |

Our run of Min-k%-Prob follows the methodology outlined in the paper; we
    run the method on one hundred 512-token spans sampled from each
    benchmark, and tune the decision threshold on a validation set of
    five of our contaminated test sets, and five test sets not used in
    our data mixture (uncontaminated). The threshold is tuned for a false
     positive rate of 5% to allow for a meaningful comparison against our
     test. A value of k=20 is used as is recommended in the paper.

We find that our method matches or exceeds the performance of this state
    of the art heuristic method, while also providing statistical proof
    of contamination.

Output:
[
    {
        "question": "Although a more efficient sharded rank comparison
            test is proposed, the computational complexity is still
            considerable. For example, testing 49 files using 1000
            permutations per shard can take 12 hours for LLaMA2.",
        "answer": "We'd like to address the concern regarding the
            computational complexity of our test. It's important to note
            that the test is a one-time process for any given model and
            dataset; once the p-values are computed, there is no need for
             recalculation. Our findings indicate that a number of
            permutations beyond 30-50 per shard offers diminishing
            returns, as shown in Figure 3 (right). Furthermore, the test'
            s design allows for easy parallelization. Each shard
            permutation can be evaluated independently, enabling the use
            of inexpensive commodity hardware to run the test
            significantly faster.",
        "is_multimodal_related": true
    },
    {
        "question": "There is no comparison with other baseline methods
            .",
        "answer": "Finally, we appreciate your suggestion to include
            baseline comparisons. We provide a comparison against a
```

```
              contamination detection method called Min-K% Prob, a state of
               the art heuristic method for contamination detection in
              language models proposed contemporaneous to our work by Shi
              et. al. (2023). We find that our method matches or exceeds
              the performance of this state of the art heuristic method.
              Please see the table in the top-level comment for numbers.
              While no other existing work is comparable in the sense that
              it provides a statistical proof of contamination like ours,
              we provide a comparison against a state of the art heuristic
              method for contamination detection called Min-K% Prob,
              proposed by Shi et. al. (2023) contemporaneous to our work.
              We use the same pretrained model and test sets from our
              experiments in Section 4.1.\n\n| Dataset     | Duplication
              Count | Sharded p (ours) | Percent Contaminated (Min-K%-Prob)
               |\n
              |------------|------------------|------------------|--------------------------
              n| BoolQ     | 1               | 0.156          | 3%
                             |\n| HellaSwag | 1               |
              0.478         | 2%                               |\n| MNLI
              | 10             | 1.96e-11         | 100%
                             |\n| MMLU-Pro-Law | 50        |  1e
              -38           | 90%                         |\n| MMLU-HS-
              Psych | 100     | 1e-38           | 74% |\n\nOur run of Min-
              k%-Prob follows the methodology outlined in the paper; we run
               the method on one hundred 512-token spans sampled from each
              benchmark, and tune the decision threshold on a validation
              set of five of our contaminated test sets, and five test sets
               not used in our data mixture (uncontaminated). The threshold
               is tuned for a false positive rate of 5% to allow for a
              meaningful comparison against our test. A value of k=20 is
              used as is recommended in the paper. We find that our method
              matches or exceeds the performance of this state of the art
              heuristic method, while also providing statistical proof of
              contamination.",
        "is_multimodal_related": false
    },
    {
        "question": "The method relies on a strong assumption of data
            exchangeability, which may not hold in real-world datasets.",
        "answer": "Regarding the assumption of data exchangeability, this
             is a strictly weaker condition than the commonly held
            assumption of independent and identically distributed (I.I.D
            .) data in machine learning. Most datasets satisfy this
            assumption to some extent.",
        "is_multimodal_related": false
    },
    {
        "question": "If a dataset is not exchangeable, how effective is
            the method?",
        "answer": "We acknowledge the validity of our test hinges on data
             exchangeability. However, depending on the source of non-
            exchangeability, it is often the case that a dataset can be
            altered slightly so that our test is still valid. For example
            , a common source of non-exchangeability is the presence of
            ascending IDs (e.g. as in SQuAD and HumanEval). We can adjust
             the d a t a by either removing these IDs or permuting the
            examples while keeping IDs c o n s t a n t to retain the test's
            applicability. This is discussed in more detail in the
            revised paper.",
        "is_multimodal_related": false
    }
]
```

## C.5 CONVERSION PROMPT

```
You are an advanced assistant trained for academic research purposes.
    Your task is to process all review-rebuttal pairs into a structured
    Question-Answer (QA) format. For every input pair, follow these
    instructions:

Input Structure:
You will process all review-rebuttal pairs, where each is provided in the
    following format:
Review: A statement or query from a reviewer providing feedback or posing
    a question about the submission.
Rebuttal: The corresponding author response addressing the feedback.

Processing Instructions:
For each review-rebuttal pair, follow the steps below in strict sequence:
1. Extract the Question (Q):
Reformulate the reviewer feedback into a clear, precise, and standalone
    question. Ensure the question:
Includes all necessary context from both the review and rebuttal (e.g.,
    clarify vague references such as "this figure" or "the results").
Is phrased in neutral and objective language, avoiding subjective or
    opinionated terms.
2. Extract the Answer (A):
Reformulate the author's rebuttal into a concise, objective, and
    standalone answer. Ensure the answer:
Directly addresses the reformulated question.
Is based strictly on the rebuttal content. Avoid additional
    interpretations, subjective language, or opinions.
3. Classify the Question:
Classify the question into a precise subcategory based on its intent
    using the schema below (see categories below).

Categories:
1. Concept Understanding [What]: Clarifies or explains key concepts,
    terminology, theoretical viewpoints, or information conveyed in
    figures, tables, or formulas.
2. Methods
    2.1. Method Disambiguation [What]: Clarifies methodological details
        to resolve misunderstandings or ambiguities, ensuring an accurate
         grasp of proposed approaches.
    2.2. Method Mechanics [How]: Questions about the implementation or
        function of methodological workflow or components, such as the
        effect of specific modules in models.
    2.3. Motivation Analysis [Why]: Examines the rationale, principles,
        or intentions underlying a proposed method or decision.
    2.4. Method Comparison : Compares the proposed approach with baseline
         methods, analyzing similarities, differences, or performance to
        highlight novelty.
3. Experiments
    3.1. Experimental Exposition [What]: Describes experimental outcomes,
         infers how modifications or variations could impact results or
        conclusions, and addresses reasoning tasks such as calculation,
        counting, or comparative analysis.
    3.2. Experimental Setup [How]: About the design, configuration, and
        execution of experiments.
    3.3. Experimental Analysis [Why]: Studies the reasons of specific
        experimental outcomes, links them to the proposed approach, and
        assesses their generalizability and potential impact.
4. Claim Verification : Binary classification tasks that assess the
    correctness of claims, hypotheses, or experimental conclusions.

Output Format: Provide the processed data for each review-rebuttal pair
    in the following JSON format:
[
```

```
    {
        "review": "Original reviewer feedback",
        "rebuttal": "Original author rebuttal",
        "Q": "Generated question",
        "A": "Generated answer",
        "Category": "Selected subcategory"
    },
    {
        "review": "Original reviewer feedback",
        "rebuttal": "Original author rebuttal",
        "Q": "Generated question",
        "A": "Generated answer",
        "Category": "Selected subcategory"
    },
    ...
]
```

## C.6 REASONING PROMPT

Open-ended QA:

```
You are an expert academic assistant. Your task is to carefully read and
    analyze the provided complete research paper, and then answer the
    following question solely based on its content, arguments, and data,
    without using any external information or assumptions.
Response Requirements:
1. The answer must be professional, precise, concise, and clearly
    presented.
2. All statements in your answer must be exclusively derived from the
    paper's content and directly relevant to the question, avoiding any
    information or claims not supported by the paper.
3. The total length of your response must not exceed 3000 characters (
    including spaces).

Question:
{question}

Paper:
{content}
```

Claim verification:

```
You are an academic judgment specialist assigned to classify the
    following statement as strictly 'True' or 'False' based exclusively
    on the content of the provided research paper. Carefully read and
    analyze the entire paper. Use only evidence directly from the text,
    and not incorporate external knowledge, assumptions, or subjective
    reasoning.

Output Requirements:
- Respond SOLELY with 'True' or 'False'
- No explanations, disclaimers, or supplementary text

Statement:
{question}

Paper:
{content}
```

## C.7 EVALUATION PROMPT

Message provided to the LLM during evaluation:

```
messages = [
    {"role": "system", "content": sys_prompt},
    {"role": "user", "content": Conciseness/Correctness/Completeness.
        format(title=title, abstract=abstract, question=question,
        reference_answer=reference_answer, predicted_answer=
        predicted_answer)},
]
```

System prompt:

```
Evaluate and rate the quality of the following predicted answer to an
    academic question according to the evaluation characteristics given
    in the system prompt.

<paper-title>{title}</paper-title>

<paper-abstract>{abstract}</paper-abstract>

<question>{question}</question>

<reference-answer>{reference_answer}</reference-answer>

<predicted-answer>{predicted_answer}</predicted-answer>
```

Conciseness:

```
<Context>
Academic question answering is the process of thoroughly reading and
    analyzing a scientific paper in order to generate answers to specific
    questions based solely on the p a p e r s content, arguments, and data
    . Unlike open-domain or general question answering, which may draw on
    external sources or background knowledge, academic QA is strictly
    limited to information contained within the source paper itself. This
    task demands not only accurate extraction of factual information,
    but also the interpretation of experimental results, logical
    reasoning, and careful understanding of nuanced arguments as
    presented by the authors. Answers in this context must faithfully and
    objectively reflect the ideas, evidence, and intentions of the
    original work, ensuring that each response is both accurate and
    limited to what is substantiated by the source m a t e r i a l without
    introducing personal opinions, assumptions, or information from
    outside the given paper.
</Context>

<Role>
You are an expert academic answer evaluator.
</Role>

<Task-Description>
The task is to evaluate the quality of a predicted answer to a given
    academic question. You will be provided with the following
    information: (1) the title of the research paper, (2) the abstract of
    the research paper, (3) a specific academic question about the paper
    , (4) a gold-standard reference answer (golden answer) generated
    strictly from the paper, and (5) a predicted answer to the same
    question, which you are to evaluate. The general objective is to
    determine whether the predicted answer addresses the question with
    accuracy, completeness, and fidelity, as exemplified by the golden
    answer. Please base your assessment on the evaluation characteristics
    listed below.
</Task-Description>

<Evaluation-Characteristics>
```

```
1. Conciseness: Evaluate whether the predicted answer is brief and to the
    point, avoiding unnecessary repetition or irrelevant information.
   The answer should deliver key content clearly, without excessive
   length or verbosity.
</Evaluation-Characteristics>

<Rating-Scale>
For each evaluation characteristic, assign a quality score between 0.00 (
   very bad) and 5.00 (very good), using decimal values precise to two
   decimal places (e.g., 3.73) for fine-grained assessment. Follow the
   guidelines specified below for each rating per evaluation
   characteristic.

1. Conciseness
0.00 1 .00 (Very bad): The predicted answer is verbose or contains
   substantial irrelevant/redundant information, making it unclear or
   unfocused.
1.01 2 .00 (Bad): The predicted answer includes some redundancy or
   unnecessary details, affecting clarity.
2.01 3 .00 (Moderate): The predicted answer is generally clear but could
    benefit from further condensation to remove several minor
   redundancies.
3.01 4 .00 (Good): The predicted answer is concise, with only minimal
   unnecessary information.
4.01 5 .00 (Very good): The predicted answer is exceptionally concise,
   presenting essential information directly and clearly with no
   redundancy.
</Rating-Scale>

<Response-Format>
For each characteristic, rate the quality with a decimal score between
   0.00 (very bad) and 5.00 (very good), precise to two decimal places (
   e.g., 4.21). Provide a short rationale for each rating.
Return your response in JSON format: {characteristic : {"rating": "", "
   rationale": ""}}

<Example-Response>
{
  "Conciseness": {
    "rating": "4.15",
    "rationale": "The answer is generally concise and focused, with only
        minimal redundant information."
  }
}
</Example-Response>
</Response-Format>

<Note>
Base your evaluation solely on the paper title, abstract, question,
   golden answer, and predicted answer provided. Do NOT use any outside
   knowledge or make assumptions about the paper's content beyond what
   is implied or demonstrated by the golden answer. Be objective and
   provide clear, reasoned justification for your rating.
</Note>
```

Correctness:

```
<Context>
Academic question answering is the process of thoroughly reading and
   analyzing a scientific paper in order to generate answers to specific
    questions based solely on the p a p e r s content, arguments, and data
   . Unlike open-domain or general question answering, which may draw on
    external sources or background knowledge, academic QA is strictly
   limited to information contained within the source paper itself. This
    task demands not only accurate extraction of factual information,
```

```
        but also the interpretation of experimental results, logical
        reasoning, and careful understanding of nuanced arguments as
        presented by the authors. Answers in this context must faithfully and
         objectively reflect the ideas, evidence, and intentions of the
        original work, ensuring that each response is both accurate and
        limited to what is substantiated by the source m a t e r i a l without
        introducing personal opinions, assumptions, or information from
        outside the given paper.
</Context>

<Role>
You are an expert academic answer evaluator.
</Role>

<Task-Description>
The task is to evaluate the quality of a predicted answer to a given
        academic question. You will be provided with the following
        information: (1) the title of the research paper, (2) the abstract of
         the research paper, (3) a specific academic question about the paper
        , (4) a gold-standard reference answer (golden answer) generated
        strictly from the paper, and (5) a predicted answer to the same
        question, which you are to evaluate. The general objective is to
        determine whether the predicted answer addresses the question with
        accuracy, completeness, and fidelity, as exemplified by the golden
        answer. Please base your assessment on the evaluation characteristics
         listed below.
</Task-Description>

<Evaluation-Characteristics>
1. Correctness: Assess the proportion of content from the reference
        answer that is accurately reflected in the predicted answer. This is
        analogous to p r e c i s i o n focus on the accuracy and fidelity of
        included information, ensuring no distortions or misrepresentations.
</Evaluation-Characteristics>

<Rating-Scale>
For each evaluation characteristic, assign a quality score between 0.00 (
        very bad) and 5.00 (very good), using decimal values precise to two
        decimal places (e.g., 3.73) for fine-grained assessment. Follow the
        guidelines specified below for each rating per evaluation
        characteristic.

1. Correctness
0.00  1  .00 (Very bad): The predicted answer consistently misrepresents
        or distorts the content of the reference answer, with substantial
        factual errors.
1.01  2  .00 (Bad): The predicted answer contains multiple inaccuracies or
         significant misinterpretations relative to the reference answer.
2.01  3  .00 (Moderate): The predicted answer accurately includes some
        content from the reference answer but may also have minor
        misstatements or factual inaccuracies.
3.01  4  .00 (Good): Most content from the reference answer is accurately
        represented in the predicted answer, with only rare errors.
4.01  5  .00 (Very good): Virtually all content from the reference answer
        present in the predicted answer is accurate and faithful, with no
        factual errors or distortions.
</Rating-Scale>

<Response-Format>
For each characteristic, rate the quality with a decimal score between
        0.00 (very bad) and 5.00 (very good), precise to two decimal places (
        e.g., 4.21). Provide a short rationale for each rating.
Return your response in JSON format: {characteristic : {"rating": "", "
        rationale": ""}}
```

```
<Example-Response>
{
  "Correctness": {
    "rating": "4.03",
    "rationale": "Most of the information in the answer accurately
        reflects the reference answer, with only minor factual
        inaccuracies."
  }
}
</Example-Response>
</Response-Format>

<Note>
Base your evaluation solely on the paper title, abstract, question,
    golden answer, and predicted answer provided. Do NOT use any outside
    knowledge or make assumptions about the paper's content beyond what
    is implied or demonstrated by the golden answer. Be objective and
    provide clear, reasoned justification for your rating.
</Note>
```

Completeness:

```
<Context>
Academic question answering is the process of thoroughly reading and
    analyzing a scientific paper in order to generate answers to specific
     questions based solely on the p a p e r s content, arguments, and data
    . Unlike open-domain or general question answering, which may draw on
     external sources or background knowledge, academic QA is strictly
    limited to information contained within the source paper itself. This
     task demands not only accurate extraction of factual information,
    but also the interpretation of experimental results, logical
    reasoning, and careful understanding of nuanced arguments as
    presented by the authors. Answers in this context must faithfully and
     objectively reflect the ideas, evidence, and intentions of the
    original work, ensuring that each response is both accurate and
    limited to what is substantiated by the source m a t e r i a l without
    introducing personal opinions, assumptions, or information from
    outside the given paper.
</Context>

<Role>
You are an expert academic answer evaluator.
</Role>

<Task-Description>
The task is to evaluate the quality of a predicted answer to a given
    academic question. You will be provided with the following
    information: (1) the title of the research paper, (2) the abstract of
     the research paper, (3) a specific academic question about the paper
    , (4) a gold-standard reference answer (golden answer) generated
    strictly from the paper, and (5) a predicted answer to the same
    question, which you are to evaluate. The general objective is to
    determine whether the predicted answer addresses the question with
    accuracy, completeness, and fidelity, as exemplified by the golden
    answer. Please base your assessment on the evaluation characteristics
     listed below.
</Task-Description>

<Evaluation-Characteristics>
1. Completeness: Assess the proportion of information in the predicted
    answer that overlaps with the reference answer. This is analogous to
     r e c a l l consider whether the predicted answer adequately covers all
    major points and details provided by the reference answer, and does
    not omit essential content.
</Evaluation-Characteristics>
```

```
<Rating-Scale>
For each evaluation characteristic, assign a quality score between 0.00 (
    very bad) and 5.00 (very good), using decimal values precise to two
    decimal places (e.g., 3.73) for fine-grained assessment. Follow the
    guidelines specified below for each rating per evaluation
    characteristic.

1. Completeness
0.00  1  .00 (Very bad): The predicted answer fails to include most of the
     key content from the reference answer, omitting essential points or
    details.
1.01  2  .00 (Bad): The predicted answer is missing several important
    aspects found in the reference answer.
2.01  3  .00 (Moderate): The predicted answer includes a moderate portion
    of the relevant content from the reference answer but lacks full
    coverage.
3.01  4  .00 (Good): Most relevant content from the reference answer is
    present, with only minor omissions.
4.01  5  .00 (Very good): The predicted answer comprehensively
    incorporates all major information from the reference answer, leaving
     out nothing significant.
</Rating-Scale>

<Response-Format>
For each characteristic, rate the quality with a decimal score between
    0.00 (very bad) and 5.00 (very good), precise to two decimal places (
    e.g., 4.21). Provide a short rationale for each rating.
Return your response in JSON format: {characteristic : {"rating": "", "
    rationale": ""}}

<Example-Response>
{
  "Completeness": {
    "rating": "3.52",
    "rationale": "The answer covers most of the key points from the
        reference answer, but omits a few minor details."
  }
}
</Example-Response>
</Response-Format>

<Note>
Base your evaluation solely on the paper title, abstract, question,
    golden answer, and predicted answer provided. Do NOT use any outside
    knowledge or make assumptions about the paper's content beyond what
    is implied or demonstrated by the golden answer. Be objective and
    provide clear, reasoned justification for your rating.
</Note>
```

