# OpenReview forum: "RPC-Bench: A Fine-grained Benchmark for Research Paper Comprehension"
_ICLR.cc/2026/Conference — ICLR 2026 Conference Withdrawn Submission_

### Official Review · Reviewer_M3V8 · 2025-10-28

**Soundness:** 2
**Presentation:** 2
**Contribution:** 2
**Rating:** 4
**Confidence:** 4

**Summary:**

This paper introduces RPC-Bench, a large-scale benchmark for evaluating fine-grained comprehension of academic papers. The dataset is constructed from real review–rebuttal exchanges sourced from OpenReview, and includes 4,050 papers and 46.3K question–answer pairs. The authors propose a taxonomy aligned with the research workflow (concepts, methods, experiments), support both textual and visual inputs, and adopt an LLM-as-a-Judge evaluation framework that jointly assesses correctness, completeness, and conciseness. Experiments across 19 models reveal significant performance gaps, especially for multimodal inputs.

**Strengths:**

1. Supporting both rendered-page images and parsed text allows fair comparison between LLMs and VLMs—a valuable contribution to document AI.

2. The nine-category fine-grained classification reflects a deep understanding of academic discourse and enables nuanced model evaluation.

**Weaknesses:**

1. The core limitation lies in using reviewer questions as comprehension targets. By nature, reviewers ask questions precisely because the paper is unclear, incomplete, or ambiguous—not because the information is present but difficult to extract. Consequently, many questions lack ground-truth answers in the original manuscript, making them unsuitable for evaluating “comprehension.” Instead, answering them often requires external knowledge, author intent inference, or speculative reasoning—tasks fundamentally different from understanding what is explicitly or implicitly stated in the paper. This undermines the benchmark’s validity as a measure of paper understanding.

2. The paper states that only the validation and test sets were manually annotated, while the training set relies on LLM-generated QA pairs. However, it provides no quantitative analysis of annotation reliability (e.g., inter-annotator agreement), no error analysis of LLM-generated questions, and no metrics on answerability (i.e., the proportion of questions actually answerable from the paper alone). The filtering criteria (“lack of substantive academic content,” etc.) are described qualitatively but lack operational definitions, raising concerns about reproducibility and dataset robustness.

3. The paper mentions filtering by citation count and acceptance status but does not clarify how these choices affect question difficulty or topical bias. More critically, it does not report how many of the final QA pairs correspond to answerable queries based solely on the paper content—a key prerequisite for any comprehension benchmark.

**Questions:**

1. What fraction of the QA pairs in the test set are answerable solely from the original paper (excluding the rebuttal)? Could the authors provide an “answerability” annotation or estimate?

2. Did the authors measure inter-annotator agreement during human validation? If so, what were the Kappa or Fleiss’ scores for category assignment and answer correctness?

3. Given that reviewer questions often target omissions or weaknesses, how do the authors justify framing RPC-Bench as a comprehension benchmark rather than a critique response or gap-filling benchmark?

---

> ### Author Response · Authors · 2025-11-26
>
> Thank you for your thorough review of our submission. We sincerely appreciate your insightful comments and constructive feedback. We have addressed each of your points in detail below:
>
> >1. Question answerability & Benchmark Objective for Paper Understanding.
>
>
> **All QA** pairs were constructed to be **fully answerable from the final camera‑ready paper**, without requiring external knowledge, inference about author intent, or unsupported assumptions. To achieve this, we:
> - Collected the authors’ final camera‑ready papers, which include clarifications, additional experiments, and supplementary content, ensuring the coverage of all relevant information needed for QA construction.
> - Removed low‑quality or unanswerable items, including those that were purely mechanical edits, relied on external resources, were indirect or non‑responsive.
> - Annotators and reviewers jointly confirmed answerability for all items. Any references to figures, formulas, or sections were checked and updated to ensure consistency with the final paper.
>
> This process ensured only questions directly supported by explicit or implicit paper content were retained. The retained QA items consist of 6.49% conceptual, 49.35% methodological, and 44.16% experimental questions, indicating that reviewers often inquire about the core concepts, methodological principles, and experimental details of a paper. The predominance of method‑ and experiment‑focused items demonstrates that answering them requires a full understanding of a paper’s core content, confirming that this benchmark is suitable for evaluating a model’s ability to understand research papers.
>
> >2. W2 & Q2
>
> - **Quantitative analysis of annotation reliability.** We assessed labeling consistency using Cohen’s Kappa. For category assignment, agreement reached 0.72 among annotators and 0.78 among reviewers; for answer correctness, 0.81 and 0.85 respectively—indicating strong consensus on task understanding and judgment.
> - **No error analysis of LLM-generated questions.** We manually checked the LLM-generated questions in the validation set and test set. Some error types include reviewers' suggestions rather than questions: (1) temporary or editorial issues, such as requests for grammar or spelling fixes, figure color/font adjustments, or formatting changes; and (2) purely mechanical requests, including adding references or open-sourcing code/data (see Appendix C.1).
> - **No metrics on answerability.** We adopted a Two‑Stage Answerability Verification to ensure all questions are answerable from the paper itself. Post‑verification analysis confirmed that ~90% of the remaining questions can be answered using only the paper’s main body, while ~10% require consulting its appendix.
> - **Explanation of "lack of substantive academic content".** “Lacking substantive academic content” referred to low‑quality QA pairs that could not be answered based on the paper. For example: 1) Relying on external sources (e.g., More cases: https://... ) 2) Giving a temporary promise instead of a real answer (e.g. Deciding what to store (or overwrite) is indeed a very interesting question that we did not explore and will address in future work.)  Detailed criteria and examples see Appendix C.1.

---

> ### Author Response · Authors · 2025-11-26
>
> >3. Weakness3: Difficulty or topical bias.
>
> We adopted three measures to ensure balanced question difficulty and topic coverage:
>
> - Papers with more than 50 citations were chosen to ensure strong academic impact. These highly cited papers are also the ones people typically want to explore in greater depth. To avoid potential bias introduced by citation count and acceptance status, we also included highly cited rejected papers and randomly sampled rejected papers. This expands coverage to emerging and less‑mainstream areas, reducing over‑representation of popular topics.
>
> - The dataset spans top computer science conferences from 2013 to 2024 across multiple subfields, capturing trends and methodological diversity. Domain distribution includes ML Theory (24.80%), Computer Vision (16.87%), NLP (15.17%), Reinforcement Learning (11.42%), Optimization (7.97%), Generative Models (6.69%), Graph ML (6.36%), AI for Science (3.23%), and other areas, yielding balanced topic distribution.
>
> - Following our taxonomy, each question is categorized from lower‑complexity “what” types (40.63%) to higher‑complexity “how” (26.56%)  and “why” types (32.81%) , covering theory (38.52%)  through applications (61.48%) . As shown in Fig. 2 on page 5, the category distribution is well balanced, indicating no significant bias in difficulty during dataset construction.
>
> >4. Question3: RPC-Bench as a comprehension benchmark.
>
> The design objectives and data construction methodology of PRC‑Bench strictly ensure that it evaluates a model’s ability to accurately understand and distill both explicit and implicit content in academic papers, rather than to speculate about missing information or to directly respond to reviewer comments.
>
> -  All data originates from the final camera‑ready versions of top‑conference papers on OpenReview. These versions resolve ambiguities and add missing details, ensuring information integrity and preventing the need for speculative “gap‑filling” responses.
>
> - Two‑stage filtering plus multi‑round human review removed questions requiring external resources or speculative inference. This ensures that the task fundamentally measures comprehension of existing content—rather than generating new content or hypothesizing about missing details.
>
> - Our four‑granularity taxonomy, with categories evenly distributed across each level, follows the natural progression of academic inquiry (what → how → why), targeting full comprehension and synthesis of existing paper content, not critique generation or missing‑content supplementation.

---

### Official Review · Reviewer_zt2U · 2025-10-29

**Soundness:** 2
**Presentation:** 2
**Contribution:** 2
**Rating:** 4
**Confidence:** 4

**Summary:**

This paper constructs a benchmark for real-world question-answering (QA) tasks using peer review data from OpenReview. In the data collection and filtering stage, a Large Language Model (LLM) was employed to help select eligible data. Finally, to ensure the validity and authenticity of the evaluation, a multi-dimensional LLM-as-a-Judge evaluation framework was proposed and aligned with human judgments. The experimental results demonstrate that this evaluation framework is consistent with human evaluation.

**Strengths:**

This paper makes significant contributions in the areas of data collection, filtering, and metric construction. Furthermore, for the proposed metrics, the authors employ a comprehensive methodology to evaluate the consistency between the LLM-as-judge and human evaluations. This process, to some extent, demonstrates the validity of the metrics.

**Weaknesses:**

Leveraging question-answering (QA) data from OpenReview to evaluate the reading comprehension capabilities of models is an intriguing approach. Compared to QA datasets generated by rule-based methods, the OpenReview data is indeed more challenging and requires a model to achieve a deeper understanding of the paper's content.

However, review questions inherently possess two characteristics that complicate their use for evaluation:

1.  **Open-endedness**: For a single question, there may be multiple reasonable answers. This ambiguity poses a significant challenge to ensuring a fair and consistent evaluation.
2.  **Information Insufficiency**: Many review questions cannot be answered by solely relying on the content of the paper. For instance, questions concerning the novelty of the work often require a broad knowledge background in the field, while specific questions about experimental details might only be answerable by the original authors.

Using such questions for model evaluation is thus inherently problematic from a fairness perspective. In fact, the evaluation of open-ended questions is a notoriously difficult problem, yet this paper dedicates limited discussion to addressing this critical issue.

**Questions:**

It would be beneficial if the authors could clarify their approach to the following challenges in dataset construction:

- The open-ended nature of review-style questions means they often lack a single ground-truth answer; responses from various perspectives may all be considered correct.
- Many review questions demand external knowledge that cannot be sourced from the primary article alone. Answers may require synthesizing information from related literature or even new experimental results. Consequently, a model confined to the input paper would naturally face significant difficulties in addressing these questions.

---

> ### Author Response · Authors · 2025-11-26
>
> We thank the reviewers for their insightful comments and constructive feedback. Our detailed responses are as follows:
> >1. Open-endedness
>
> Although a question may have multiple reasonable answers, we prioritized objective questions during annotation to ensure more definitive answers. In addition, we used the authors’ responses—those who best understand the paper—as the reference answers, which guarantees their authoritativeness.
>
> To evaluate multiple semantically-equivalent answers, we draw inspiration from recent LLM-as-Judge research and propose a standardized framework assessing three core dimensions (Conciseness, Correctness, and Completeness) by comparing predicted answers with ground truth to assess their semantic consistency.
>
> Our evaluation framework is consistent, scalable, and addresses the limitations of traditional semantic similarity metrics (e.g., BLEU, BERTScore), which may fail when multiple semantically equivalent answers exist (Section 4.2).
>
> >2. Information Insufficiency
>
> We curated the dataset to ensure all questions are fully answerable using only the source paper. We relied on the final camera-ready versions submitted by authors, which often include clarifications, additional experiments, and refined novelty claims.
>
> A two-stage filtering (review–rebuttal filtering and QA filtering) removed low-quality or unanswerable items, such as reviewers' edit requests (e.g. adding references, fixing grammar errors), responses requiring external resources, or giving a temporary promise instead of a real answer, etc.
> - GLM‑4‑Plus screened all review–rebuttal pairs and QA items, removing 24.5% of pairs and 12.96% of items as low‑quality or unanswerable.
> - Annotators and reviewers jointly removed an additional 8.87% upon direct inspection.
>
> During annotation, both annotators and reviewers verified question answerability. References to figures, formulas, or sections were confirmed in the final paper, and indices were updated accordingly.

---

> > ### Comment · Reviewer_zt2U · 2025-11-27
> >
> > Thank you for your responses. In your first response, you mentioned using author responses as the reference answers because authors are the most knowledgeable about the paper. However, this is precisely the problem. Authors know a great deal of information may not explicitly stated in the paper. How can you guarantee that the model can answer these questions after reading the paper?
> >
> > Furthermore, you mentioned focusing more on objective questions. How is this objectivity assessed? And if we're evaluating objective questions, what advantages do you have compared to existing multi-hop QA datasets (such as hotpotQA, Musique, etc.)?

---

> > > ### Author Response · Authors · 2025-11-28
> > >
> > > Thank you for your response.
> > >
> > > Our work is designed to evaluate a model’s ability to comprehend research papers, which inherently requires the model to possess relevant domain knowledge in order to interpret the concepts, methods, and experiments described in scholarly articles. Figure 3(a) shows that GPT attains F1‑like scores above 70% for both concept understanding and method disambiguation, indicating that recent LLMs are capable of grasping foundational domain concepts that authors may omit in their papers.
> > >
> > > To ensure all evaluation questions are answerable from the paper content (see 2: Information Insufficiency), we: (1) use the final camera‑ready versions of papers, (2) apply a two‑stage filtering process and (3) conduct human expert verification. Human experts are required to check whether authors' responses exist in the original paper.
> > >
> > > Our taxonomy follows the natural progression of academic inquiry (what → how → why) and focuses exclusively on the objective understanding of academic papers. During annotation, we strictly adhered to this taxonomy and removed any out-of-scope, non-objective questions such as:
> > >
> > > - “Would it be possible to integrate evaluation metrics with more complex prompts (e.g., more spatial grounding) to collect human feedback?"
> > >
> > > - “It is not easy to collect large-scale human preferences for reward model training. Is there a better way to construct an Image Reward model without fully relying on human annotations?”
> > >
> > > It is also important to note that our work differs substantially from datasets such as HotpotQA and MuSiQue:
> > >
> > > 1. **Task focus:** We target the understanding of academic papers, whereas HotpotQA and MuSiQue focus on multi-hop reasoning tasks.
> > >
> > > 2. **QA construction:** We decompose and rewrite review–rebuttal pairs into QAs using our taxonomy, while HotpotQA builds a Wikipedia hyperlink graph and MuSiQue constructs QAs by combining multiple single-hop questions.
> > >
> > > 3. **Answer length:** Our open-ended QA answers are significantly longer, **averaging 90.9 words**, reflecting the complexity of academic paper content.
> > >
> > > 4. **Evaluation methodology:** We evaluate answers using multiple LLM as judges that closely align with human assessments, scoring on conciseness, correctness, and coverage. In contrast, HotpotQA and MuSiQue rely on traditional metrics such as exact match (EM) and F1.

---

### Official Review · Reviewer_yz9y · 2025-10-31

**Soundness:** 3
**Presentation:** 4
**Contribution:** 3
**Rating:** 6
**Confidence:** 3

**Summary:**

The paper introduces **RPC-Bench**, a new large-scale, fine-grained benchmark designed to evaluate the comprehension capabilities of Large Language Models (LLMs) and Visual Language Models (VLMs) on research papers. The authors identify that existing benchmarks are often limited in scale, rely on synthetic questions, or lack nuanced evaluation metrics.

To address this, RPC-Bench is constructed from **4,050 high-quality academic papers** and their **46.3K authentic review-rebuttal exchanges** sourced from OpenReview. This real-world data provides complex, expert-level questions. The benchmark supports two input formats: pure text (Markdown) and rendered page images, enabling the evaluation of both LLMs and VLMs.

A key contribution is the paper's **fine-grained taxonomy**, which categorizes questions based on the research workflow (Concepts, Methods, and Experiments) and their intent (what, how, why). Data was generated using a collaborative **LLM-human framework**, where LLMs decomposed and rewrote review-rebuttal pairs into a QA format, and human annotators (Master's level or higher) validated and refined the test and validation sets.

For evaluation, the paper proposes a scalable "LLM-as-a-Judge" protocol that measures three dimensions: **correctness**, **completeness**, and **conciseness**. These are combined into an "F1-like" score (harmonic mean of correctness and completeness) and an "Informativeness" score (F1-like penalized by lack of conciseness).

Experiments on 19 models (LLMs, VLMs, DCMs, and RAG) reveal significant limitations in current SOTA systems. The best model, GPT-5, achieved a 66.54% F1-like score, which dropped to 35.05% on the conciseness-penalized "Informativeness" metric. A major finding is that multimodal models consistently performed worse with visual-text inputs than with text-only inputs, highlighting a critical gap in visual reasoning for scholarly documents.

**Strengths:**

1.  **Authentic and Realistic Data Source:** The benchmark's foundation in real peer review-rebuttal exchanges from OpenReview is a major strength. This moves beyond synthetic QA and captures the genuine, nuanced, and complex questions that domain experts ask, providing a more challenging and realistic test of comprehension.
2.  **Novel and Robust Evaluation Protocol:** The paper thoughtfully moves beyond simplistic metrics like ROUGE or BERTScore, which it demonstrates are insufficient for this task. The proposed "LLM-as-a-Judge" framework with its multi-dimensional metrics (correctness, completeness, conciseness) offers a much more nuanced and semantically meaningful assessment.
3. **Validation of the Evaluation Framework:** The authors validate their LLM-as-a-Judge protocol by comparing it against human preferences on a 300-instance sample. The high agreement found (e.g., ~0.85 average correlation for the GPT-5 judge) builds significant trust in the benchmark's results.
4. **Dual-Modality Support:** The inclusion of both pure-text and rendered-page (image) inputs is a significant contribution. It allows the benchmark to evaluate and compare LLMs and VLMs directly, leading to the important finding that current multimodal models struggle to effectively integrate visual information from scholarly documents.
5. **Fine-Grained Taxonomy:** The taxonomy, structured around the research flow (Concepts, Methods, Experiments) and "what/how/why" questions, is highly logical. This fine-grained categorization enables detailed error analysis, revealing *where* models fail (e.g., performing better on "Concept Understanding" than on "Experimental Analysis").

6.  **Scalable Annotation Pipeline:** The "LLM-human interaction annotation framework" is a practical and scalable approach. Using powerful LLMs (like GPT-40) for initial data decomposition and other LLMs for rewriting, followed by human validation and refinement, balances scalability with quality control.

**Weaknesses:**

1. **Unverified Training Data:** The paper explicitly states that "only the validation and test sets were manually annotated, while the training set retained QA pairs generated by LLMs". This means the vast majority of the QA pairs (39,203 for training vs. 6,152 for val and 2,787 for test) are of unverified, and likely lower, quality. This could negatively impact any models fine-tuned on this data, potentially teaching them to replicate LLM-generated artifacts.
2. **Potential Domain Bias:** The data is sourced exclusively from OpenReview. While this is a high-quality source, it is heavily dominated by Computer Science and related fields (like AI/ML). The paper does not analyze or discuss the domain distribution, so the benchmark's applicability to research papers from other domains (e.g., life sciences, humanities, social sciences) is unclear. Moreover, not all reviews are published, so the potential selection bias is not mentioned nor mitigated in this case.
3. **Artificial Input Constraints:** The methodology involves practical but significant truncations. Text inputs are "truncated if it exceeds the model's context window", and image-based inputs are limited to the "first 15 pages". This means the benchmark does not fully test whole-document comprehension, as relevant information may appear in appendices or beyond the 15-page or context-window limit.
4. **Ambiguity in the "Informativeness" Metric:** The "Informativeness" score is defined as $F1\text{-like} \times (\text{Conciseness} / 5)$, which heavily penalizes verbosity. However, the paper's own analysis notes that models sometimes produce "overly long outputs" that consist of "repetitive, non-informative text" as a *failure mode*. The finetuning analysis also shows conciseness is more easily learned than correctness or completeness. This suggests that heavily coupling the F1-like score with conciseness might be an overly aggressive penalty that conflates

**Questions:**

**Questions for the Authors:**

1.  **On the Quality of the Training Data:** You state that only the validation and test sets were manually annotated, while the 39.2K pairs in the training set consist of LLM-generated data. What analysis was performed to validate the quality of this large, unverified training set? How can we be sure that models fine-tuned on this data are not simply learning to replicate the specific generative artifacts or failure modes of the LLMs (GLM-4-Plus and DeepSeek-V3) used to create it, rather than learning true research comprehension?

2.  **Rationale for the "Informativeness" Metric:** The "Informativeness" metric is defined as $F1\text{-like} \times (\text{Conciseness} / 5)$, which results in a significant score drop (e.g., GPT-5 from 66.54% to 35.05%). Your fine-tuning analysis suggests conciseness is "more readily learn[ed]" than correctness or completeness. Why did you choose to penalize the primary F1-like score so heavily with conciseness, rather than reporting it as a separate, complementary metric? Does this not risk conflating stylistic adherence with core comprehension?

3.  **Impact of Input Truncation:** You note that text inputs are truncated if they exceed the context window and image inputs are limited to the first 15 pages. How frequently did this truncation occur for the models tested? How can this benchmark claim to test full-paper comprehension if the model may be prevented from accessing relevant information located in appendices or later sections of the paper, which are often critical for answering detailed experimental or methodological questions?

5.  **Interpreting the VLM Performance Drop:** A key finding is that VLMs perform significantly *worse* with multimodal inputs than with text-only inputs. This is counter-intuitive, as figures and tables should provide *additional* grounding. Your own case study (Example 2) highlights the necessity of multimodal grounding. What is your hypothesis for this performance collapse? Is it a failure of the models to parse rendered text, an inability to integrate visual information with long-range textual context, or an artifact of the 15-page limit on visual inputs?

---

> ### Author Response · Authors · 2025-11-26
>
> Thank you for your thorough review of our submission. We sincerely appreciate the insightful comments and constructive feedback provided. Below, we address each of your points in detail:
>
> >1. Unverified Training Data, On the Quality of the Training Data
>
> Compared to the dataset at the time of paper submission, we further annotated 435 papers, yielding 6,827 QA pairs. **This expansion increased the validation set to twice the size of the original dev set (i.e., 12,979 QA pairs in validation set).** This enables the construction of high-quality validation and test sets with verified annotations, and the currently annotated QA pairs are already sufficient for supervised fine-tuning. In addition, the unverified training set can be retained as an optional resource for researchers.
>
> For the training set, we employed LLMs to convert the original review-rebuttal exchanges into multiple QA pairs. The models served solely as format converters and did not introduce any additional information. Subsequently, a two-stage filtering process removed 37.46% of the items that failed to meet our quality criteria (e.g., reviewer requests to add references or QA pairs requiring external knowledge).
>
> To validate training data quality, we fine‑tuned the Qwen3‑4B‑Instruct‑2507 model on: (1) Training set only; (2) New validation set only; (3) Combined training + validation sets
>
> | Model               | Conc.  | Corr.  | Compl. | F1-like | Info. |
> |---------------------|--------|--------|--------|---------|-------|
> | Qwen3-4B            | 29.15  | 25.06  | 23.72  | 24.37   | 7.11  |
> | Qwen3-4B-train      | 74.83  | 35.22  | 27.28  | 30.74   | 23.01 |
> | Qwen3-4B-dev        | 66.06  | 37.34  | 30.52  | 33.59   | 22.19 |
> | Qwen3-4B-train+dev  | 70.71  | 37.75  | 30.27  | 33.60   | 23.76 |
>
> From the table, we observe that the model fine-tuned on the training set outperforms the model without fine-tuning. Furthermore, the model fine-tuned on both the training and validation sets yields better results in terms of Informativeness than the model trained solely on the validation set. This validates the quality of the training data, demonstrating its ability to enhance the model's response performance on the test set.
>
> >2. Potential Domain Bias
>
> Our benchmark currently covers computer science and its subfields, with the following distribution:
>
> ML Theory (24.80%), Computer Vision (16.87%), NLP (15.17%), Reinforcement Learning (11.42%), Optimization (7.97%), Generative Models (6.69%), Graph ML (6.36%), AI for Science (3.23%), and other areas, yielding balanced topic distribution.
>
> During dataset construction, We prioritized papers with more than 50 citations, as higher citation counts are generally associated with stronger research contributions and broader scholarly recognition, making them more suitable for in‑depth comprehension in the QA context.
>
> To avoid over‑representing popular topics from highly‑cited accepted papers, we additionally incorporated high‑citation rejected papers as well as randomly sampled rejected papers. This approach captures works of varying maturity levels and includes non‑mainstream research directions, ensuring diversity in both methodology and topic coverage to avoid potential bias.
>
> >3. Artificial Input Constraints, Impact of Input Truncation
>
> Limitations such as text truncation and image constraints during inference come from current LLM/VLM capabilities, not from our benchmark design.
>
> - Our dataset contains the complete text and images. If a paper fits within a model’s input limit, we provide it in full; otherwise, truncation is applied.
>
> - Most modern LLMs can handle ≥ 128K tokens, enabling full‑paper processing in most cases, except for a few extremely long papers.
>
> - For VLMs, input limits depend on image resolution, image count, GPU memory, and overall context size. As academic papers typically have ≤ 10 pages in the main body, we provide the first 15 pages as visual context.
>
> - Only a small proportion of questions require appendix‑specific content to produce a complete answer: dev = 11.04%, test = 10.08%.

---

> ### Author Response · Authors · 2025-11-26
>
> >4. Ambiguity in the "Informativeness" Metric, Rationale for the "Informativeness" Metric
>
> Our evaluation framework measures Conciseness, Correctness, and Completeness, supplemented by F1‑like and Informativeness. The Informativeness metric integrates correctness, completeness, and brevity, offering a balanced view of paper comprehension quality.
>
> - Conciseness is not merely a stylistic preference; it is a crucial quality factor that reduces redundancy and semantic noise.
>
> - Separate dimension assessment may overlook interaction effects; informativeness captures overall performance in a holistic way that prevents inflated scores from verbose outputs.
>
> - For example, assuming a ground-truth answer is “A, B, C, D, E,” a candidate answer is “A, A, A, B, B, B, C, C, C, D, D, D.” This candidate answer achieves 100% precision, 80% recall, and 0.89% F1. However, its output is verbose. This answer should be penalized in terms of conciseness.
>
> >5. Interpreting the VLM Performance Drop
>
> To further investigate the observed performance gap, we conducted a case study analysis (Appendix B.4). Cases 1, 2, and 4 shows that LLMs consistently deliver higher factual accuracy, greater answer completeness, and closer alignment with reference answers than VLMs.
>
> In Case 3, VLMs accurately identified methodological boundaries (e.g., excluding tasks outside the intended domain‑specific RAG scope) and maintained reasoning consistency with the reference. This indicates that VLMs can effectively capture high‑level conceptual scope, especially when the task focuses on boundary recognition rather than detailed fact retrieval.
>
> Overall, LLMs excel in tasks requiring precise, evidence‑linked, detail‑rich answers from textual content. VLMs occasionally match or surpass LLMs in identifying conceptual scope from visual inputs. The observed gap largely reflects VLMs’ difficulty in extracting high‑precision information from PDF‑rendered images of dense academic content (e.g., complex tables, multi‑column layouts, long texts), which often results in incomplete or noisy answers compared with clean text inputs.
>
> VLMs’ practical capacity is limited by image resolution, number of images, GPU memory, and overall context length. Since typical papers’ main bodies rarely exceed 10 pages, we provide the first 15 pages as visual context to ensure coverage of core content without exceeding model limits. This approach balances coverage and efficiency.

---

### Official Review · Reviewer_W4at · 2025-11-01

**Soundness:** 3
**Presentation:** 2
**Contribution:** 2
**Rating:** 4
**Confidence:** 3

**Summary:**

This paper introduces RPC-Bench, a fine-grained benchmark for evaluating LLMs and MLLMs in research paper comprehension. The dataset is built from OpenReview review–rebuttal interactions and covers nine reasoning categories reflecting different aspects of research understanding. This paper also propose an LLM-as-a-Judge evaluation framework based on correctness-completeness and conciseness.

**Strengths:**

1. The paper uses authentic and reliable data sources, constructing the benchmark from OpenReview review–rebuttal dialogues rather than synthetic data, and filtering for high-quality papers.
2. It defines a nine-category framework that comprehensively covers multiple cognitive dimensions of research understanding.
3. Beyond traditional evaluation metrics, this paper introduces a three-dimensional assessment combining correctness, completeness, and conciseness to better capture model performance.

**Weaknesses:**

1. The experiment analysis is not sufficiently in-depth. Although the benchmark defines nine categories, the experiments only report scores and brief observations without detailed analysis of each category. Moreover, the paper compares LLMs and VLMs but does not analyze why RAG-based methods fail to show advantages.
2. The motivation emphasizes that research paper comprehension requires domain knowledge and deep reasoning, and the taxonomy covers methodological and analytical capabilities. However, the experiments do not isolate or quantify the effect of domain-specific knowledge, making it unclear whether performance gaps stem from reasoning difficulty or knowledge limitations.
3. The case study is too shallow. It lacks representative failure examples and does not illustrate model errors in different resources. More interpretable examples of LLM-as-a-Judge scoring would strengthen the analysis beyond the simple prompt examples in the appendix.

**Questions:**

Please refer to the Weakness. Additionally,
1. Can the authors provide statistics on the field diversity of papers included in RPC-Bench?
2. Are there any plans to expand RPC-Bench beyond computer science papers, or to include cross-disciplinary samples to test generalization of research comprehension?

---

> ### Author Response · Authors · 2025-11-26
>
> We are sincerely thankful for your detailed review and the insightful feedback you have offered. These comments have been extremely helpful in improving our manuscript, and we respond to each issue in detail below.
>
> >1. W1: ... Detailed analysis of each category. ...
>
> Our evaluation across eight open-ended QA categories reveals clear differences w.r.t. Question types and model types:
>
> - Concept Understanding & Disambiguation Questions: Require accurate in-depth understanding. High‑capacity LLMs integrate contextual details effectively. VLMs often omit fine‑grained elements; Document-Centric Models (DCMs)' answers tend to be too general. RAG excels at information extraction but lacks summarization and reasoning capabilities.
>
> - Method Mechanics & Comparison: LLMs provide clear explanatory contrasts. RAG provides basic comprehension but offers less interpretive depth. VLMs may miss details; DCMs risk repetitive or partial answers.
>
> - Method Motivation & Experimental Setup: Focus on understanding the rationale behind methodological/experimental choices. LLMs and strong VLMs link design choices to their underlying motivation; DCMs and RAG often miss underlying rationale.
>
> - Experiment Exposition & Analysis: Demand retrieval from dispersed sources, sometimes with quantitative evidence. LLMs can perform qualitative or quantitative analysis of experimental results based on a research paper; VLMs further perform analyses via figure interpretation; RAG ensures detail fidelity. DCMs' answers tend to be too general.
>
> Overall, LLMs deliver the most complete and context‑aware answers, collecting dispersed information, and capturing nuanced contrasts. VLMs improve detail coverage when visual data is relevant, but sometimes struggle with reasoning over long contexts. DCMs show high variance and their answers tend to be too general without locating relevant parts. RAG achieves high‑precision fact retrieval but lacks in-depth understanding and reasoning capabilities.
>
> >2. Weakness1: ... but does not analyze why RAG-based methods fail to show advantages ...
>
> We conducted a further analysis of the RAG-based approach, focusing on representative cases (Appendix B.4):
>
> - Case 1: The model successfully retrieved information related to the question but failed to use it effectively in answer generation. Key links between the paper’s context and the question were overlooked, and model‑generated hallucinations were introduced.
>
> - Case 2: The model failed to retrieve the key information needed to answer the question, such as the main innovations of the method. Without this critical material, the generated answer was incomplete or incorrect.
>
> - Case 3: The model broke the question down into sub‑questions, retrieved all necessary content, and produced a coherent, complete answer.
>
> Overall, The main bottleneck of current RAG‑based approach lies in their limited ability to accurately understand complex questions, retrieve targeted knowledge, and integrate information from multiple sources into a unified, meaningful answer.

---

> ### Author Response · Authors · 2025-11-26
>
> >3. Weakness2: ... the experiments do not isolate or quantify the effect of domain-specific knowledge, making it unclear whether performance gaps stem from reasoning difficulty or knowledge limitations.
> Question1: Can the authors provide statistics on the field diversity of papers included in RPC-Bench?
>
> The current version of our benchmark primarily covers computer science and its subfields, with the following domain distribution:
>
> | Field                       | F1-like  | Proportion   |
> |-----------------------------|--------|-----------|
> | Machine Learning Theory     | 52.37  | 24.8%     |
> | Optimization                | 51.19  | 7.97%     |
> | Reinforcement Learning      | 69.11  | 11.42%    |
> | Graph Machine Learning      | 77.32  | 6.36%     |
> | Generative Models           | 56.32  | 6.69%     |
> | Computer Vision             | 53.42  | 16.87%    |
> | Natural Language Processing | 56.28  | 15.17%    |
> | Speech and Audio Processing | 39.81  | 1.35%     |
> | Robotics                    | 35.23  | 1.92%     |
> | Multimodal Learning         | 58.28  | 2.33%     |
> | AI for Science              | 87.64  | 3.23%     |
> | Other                       | 47.37  | 1.89%     |
>
> Based on our taxonomy, what-type questions generally require solid domain knowledge, while how and why-type questions demand deeper reasoning and analytical skills based on the paper. As shown in Fig. 3(a) and the table above, model performance declines as questions move from surface‑level to deeper inquiry, and as topics progress from concepts to methods to experiments, spanning from theoretical to applied domains. For example, the best-performing model (GPT, F1‑like) drops from 80.35% on Concept Understanding to 64.59% on  Experimental Analysis. This indicates that current models possess a reasonable level of domain knowledge but struggle to capture finer‑grained details and are less adept at uncovering the underlying principles behind methods and experiments.
>
> We evaluated the QA performance on theory-oriented and application-oriented papers and found no significant difference between them (p-value = 0.77). This suggests that the bottleneck in understanding papers may lie primarily in comprehension and reasoning about the paper itself, rather than in domain-specific knowledge.
>
> >4. Weakness3: The case study is too shallow ...
>
> We have expanded the case study in three key aspects: (1) detailed analysis of each category, (2) evaluation of the RAG approach, and (3) comparison of LLM and VLM performance. Full results and discussion are provided in Section 4.5 and Appendix B.4.
>
> >5. Question2: Are there any plans to expand RPC-Bench ...
>
> We plan to expand in stages: (1) Add coverage for life sciences and social sciences, (2) Gradually extend to additional disciplines, improving the benchmark’s generality and representativeness.
>
> Recent Progress:
>
> We have already enlarged the annotation scale within the existing CS domain: 435 new papers have been annotated, generating 6,827 QA samples. This expanded validation set is twice the size of the original dev set **(i.e., 12,979 QA pairs in validation set)**, providing a stronger data foundation for cross‑domain expansion.
>
> Next Steps:
>
> - Incorporate publicly available peer‑review resources from other fields, e.g., PeerJ (ecology, biology, interdisciplinary humanities) and F1000Research (life sciences, medicine, social sciences).
>
> - Apply our established construction and filtering pipeline to ensure consistent quality and answerability.
>
> - Use domain quotas and time‑span controls to avoid topic bias or temporal bias in any non‑CS domain.
>
> - Adapt the QA annotation taxonomy to account for differences in writing and reviewing styles across disciplines, ensuring cross‑domain comparability of question types.

---

### Note · Authors · 2026-01-05

I have read and agree with the venue's withdrawal policy on behalf of myself and my co-authors.